# InvFusion: Bridging Supervised and Zero-shot Diffusion for Inverse Problems

**Noam Elata**[*]
Technion
Haifa, Israel
noamelata@campus.technion.ac.il

**Hyungjin Chung**[*]
EverEx
Seoul, South Korea
hj.chung@everex.ac.kr

**Jong Chul Ye**
KAIST
Daejeon, South Korea
jong.ye@kaist.ac.kr

**Tomer Michaeli**
Technion
Haifa, Israel
tomer.m@ee.technion.ac.il

**Michael Elad**
Technion
Haifa, Israel
elad@cs.technion.ac.il

## Abstract

Diffusion Models have demonstrated remarkable capabilities in handling inverse problems, offering high-quality posterior-sampling-based solutions. Despite significant advances, a fundamental trade-off persists regarding the way the conditioned synthesis is employed: Zero-shot approaches can accommodate any linear degradation but rely on approximations that reduce accuracy. In contrast, training-based methods model the posterior correctly, but cannot adapt to the degradation at test-time. Here we introduce InvFusion, the first training-based degradation-aware posterior sampler. InvFusion combines the best of both worlds – the strong performance of supervised approaches and the flexibility of zero-shot methods. This is achieved through a novel architectural design that seamlessly integrates the degradation operator directly into the diffusion denoiser. We compare InvFusion against existing general-purpose posterior samplers, both degradation-aware zero-shot techniques and blind training-based methods. Experiments on the FFHQ and ImageNet datasets demonstrate state-of-the-art performance. Beyond posterior sampling, we further demonstrate the applicability of our architecture, operating as a general Minimum Mean Square Error predictor, and as a Neural Posterior Principal Component estimator.[1]

## 1   Introduction

Diffusion models [49, 52, 28] have emerged as a leading class of generative machine learning techniques [16, 44]. Since their inception, diffusion models have gained significant traction in solving complex inverse problems, such as super-resolution [47] or in-painting [37], where the goal is to reconstruct or estimate an underlying image from partial or degraded observations.

Diffusion models can be trained to solve inverse problems using a simple conditioning framework, in which the measurements are supplied to the network [46, 47]. Although powerful, these training-based methods are typically restricted to handling a limited set of degradations per trained network, as they must learn the connection between clean images and their degraded versions. Moreover, existing training-based models cannot accept the degradation as an input and must rely on the measurements provided at test-time to infer which among the possible degradations to restore. This forces a blind

---

[*]Equal Contribution
[1]Code implementation available at https://github.com/noamelata/InvFusion

39th Conference on Neural Information Processing Systems (NeurIPS 2025).

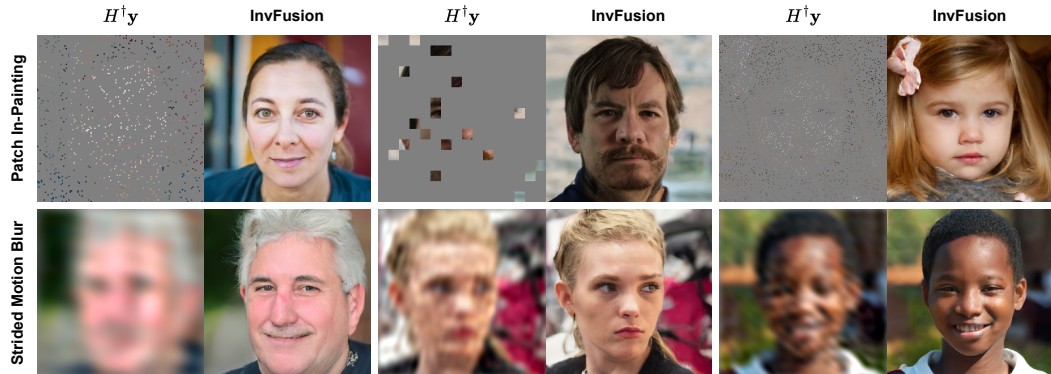

Figure 1: **Examples for posterior samples from our degradation-aware diffusion model.** A single model can restore multiple degradations, such as in-painting, de-blurring and super-resolution with high image fidelity and realism, by integrating the degradation operator into the model's architecture.

restoration setting, which is inaccurate when multiple degradations fit the measurements. These fundamental limitations significantly reduce the applicability of trained conditional diffusion-based methods, as no single model can accommodate a wide range of inverse problems. Conversely, zero-shot methods [10, 11, 32, 33, 51] utilize unconditional pre-trained diffusion models for solving inverse problems, taking into account the particular degradation process from which each input suffers, enabling a more precise non-blind setting. They are thus remarkably flexible in handling various degradation scenarios. However, these approaches use approximation that reduce accuracy and suffer from computational inefficiency.

Here, we introduce InvFusion, an architectural framework that combines the strengths of training-based and zero-shot methodologies. Our approach is the first training-based method that can be informed of the degradation at test-time. The InvFusion architecture is conditioned on the degradation operator, enabling the network to make full use of the provided measurements. Since the set of all possible degradations is too large to be used directly as a model input, we propose integrating the degradation operator into the architecture's design, implicitly enabling the network to infer whether its intermediate features correspond to the measurements. This novel architecture allows the network to adapt to a wide range of degradations while simultaneously maintaining the high performance of training-based networks. Our method represents a paradigm shift, offering a more versatile, accurate, and computationally efficient approach to handling complex inverse problems.

Our network's architecture is inspired by unrolling frameworks [54, 61, 1], in which the layers mimic an optimization process. In our case, we integrate the degradation operator within the network's layers using attention mechanisms [57, 17, 43] . The attention layers preserve dimensionality, and thus enable applying the degradation operator to all network activations. We use HDiT [13] as our base architecture, which is particularly effective in facilitating operations on higher-resolution images, as shown in Fig. 1.

Accordingly, our method bypasses the approximations of zero-shot methods and solves the degradation-ambiguity of existing training-based method, offering significant advantages in accuracy, computational efficiency, and flexibility. In particular, our method can be used with any sampler designed for unconditional diffusion models [35, 36, 30], including Classifier-Free Guidance (CFG) [27, 12] and related techniques.

Our experiments on FFHQ [29] and ImageNet [15] evaluate various approaches for addressing multiple restoration tasks with a single trained model. InvFusion out-performs existing inverse problem solvers, both in the training-based and zero-shot categories, establishing a new state-of-the-art (SOTA). Beyond posterior sampling, we illustrate how our architecture can also be used for Minimum Mean Square Error (MMSE) prediction, as well as for training a Neural Posterior Principal Component (NPPC) [42] predictor, enabling uncertainty quantification for a wide variety of restoration tasks with a single model.

## 2 Background and Related Work

### 2.1 Diffusion Models

Diffusion Models [49, 52, 28] generate high-quality images through a sequential Gaussian noise removal process. Given a source data distribution $p(\mathbf{x}_0)$, a forward diffusion process constructs a Stochastic Differential Equation (SDE) $d\mathbf{x}_t = -\frac{1}{2}\beta(t)\mathbf{x}_t + \sqrt{\beta(t)}d\mathbf{w}_t$, where $\mathbf{w}_t$ is a standard Wiener process and $\beta(t)$ is a deterministic function of $t \in [0, 1]$ taken such that at $t = 1$, the final marginal distribution $p(\mathbf{x}_1)$ becomes approximately a standard Gaussian. Each marginal distribution along this SDE, $p(\mathbf{x}_t)$, can be constructed directly by adding Gaussian noise to the source data distribution, $\mathbf{x}_t = \sqrt{\bar{\alpha}(t)}\mathbf{x}_0 + \sqrt{1 - \bar{\alpha}(t)}\epsilon$, where $\bar{\alpha}(t) = e^{-\int_0^t \beta(s)ds}$ and $\epsilon$ is a standard Gaussian random vector. To generate a sample image within this framework, one starts with a sample of white Gaussian noise and employs an appropriate SDE solver [35, 36, 30] for the reverse SDE [4, 39, 53]. The latter requires knowing the score $\nabla_{\mathbf{x}_t} \log(p(\mathbf{x}_t))$ [52, 50, 53]. By exploiting the connection between the score and the MMSE predictor [58, 2], $\mathbf{x}_t + (1 - \bar{\alpha}(t))\nabla_{\mathbf{x}_t} \log(p(\mathbf{x}_t)) = \sqrt{\bar{\alpha}(t)}\mathbb{E}[\mathbf{x}_0|\mathbf{x}_t]$, a denoising network trained to approximate $\mathbb{E}[\mathbf{x}_0|\mathbf{x}_t]$ can be used in place of the real score function. Training such a network is done using a regression loss,

$$\mathcal{L} = \mathbb{E}\left[w_t \|m_\theta(\mathbf{x}_t, t) - \mathbf{x}_0\|^2\right], \tag{1}$$

where $m_\theta$ is the model being trained, $\mathbf{x}_0 \sim p(\mathbf{x}_0)$, and $w_t$ is some weighting function.

### 2.2 Inverse Problems

Inverse problem solvers attempt to reverse a degradation process that corrupted an image $\mathbf{x} \in \mathbb{R}^D$ and yielded measurements $\mathbf{y} \in \mathbb{R}^d$. Many common degradations, such as those encountered in the super-resolution, deblurring, denoising, and in-painting tasks, have a linear form. These degradations are often written as $\mathbf{y} = \boldsymbol{H}\mathbf{x} + \mathbf{n}$ where $\boldsymbol{H} \in \mathbb{R}^{d \times D}$ and $\mathbf{n} \sim \mathcal{N}(0, \sigma_n^2 I)$ is white Gaussian noise added to the measurements. Noiseless measurements can be formulated with $\sigma_n = 0$. When attempting to design a system that can handle multiple degradations, $\boldsymbol{H}$ can be considered a random matrix drawn from some distribution of possible corruption operators.

A popular approach for generating a reconstruction $\hat{\mathbf{x}}$ of $\mathbf{x}$ based on the measurement $\mathbf{y}$ and on knowledge of $\boldsymbol{H}$, is to draw $\hat{\mathbf{x}}$ from the posterior distribution, $\hat{\mathbf{x}}_{\text{Post}} \sim p(\mathbf{x}|\mathbf{y}, \boldsymbol{H})$. Methods aiming for this solution are called posterior samplers and are the main focus of this work. Another popular approach to generating a reconstruction, is aiming for the Minimum Mean Square Error (MMSE) predictor, $\hat{\mathbf{x}}_{\text{MMSE}} = \mathbb{E}[\mathbf{x}|\mathbf{y}, \boldsymbol{H}]$, which, as its name implies, achieves the lowest possible squared-error distortion. Obtaining an approximation of the MMSE predictor can be done by training a regression network $m_\theta(\mathbf{y})$ to minimize $\mathcal{L} = \mathbb{E}[\|m_\theta(\mathbf{y}) - \mathbf{x}\|^2]$.

When the degradation operator $\boldsymbol{H}$ is not known, or is known but cannot be provided to the model, the relevant posterior distribution is

$$p(\mathbf{x}|\mathbf{y}) = \int p(\mathbf{x}|\mathbf{y}, \boldsymbol{H})p(\boldsymbol{H}|\mathbf{y})d\boldsymbol{H}. \tag{2}$$

This posterior is a weighted average of all possible posterior distributions (for all possible degradations $\boldsymbol{H}$), each weighted by the corresponding $p(\boldsymbol{H}|\mathbf{y})$. This posterior may generally encompass a significantly larger uncertainty regarding $\mathbf{x}$. One key limitation of training-based posterior samplers (like Palette [46]) is that they currently lack a mechanism to condition the model on $\boldsymbol{H}$. Therefore, even if $\boldsymbol{H}$ is known, it cannot be provided to the model at test time, so that the model is unavoidably tasked with solving a blind restoration problem. In this work, we introduce a *degradation-aware* model architecture, enabling the model to incorporate information about the degradation $\boldsymbol{H}$. This allows solving the precise non-blind restoration task whenever the degradation is known at test time.

### 2.3 Diffusion Restoration

In recent years, diffusion models have become the leading approach for posterior sampling. By modifying the probability distribution into a conditional one, $p(\mathbf{x}_t|\mathbf{y})$, solutions to the inverse problem can be sampled using the score function of these conditional distributions. The most straightforward

approach to leverage diffusion models for posterior sampling is to alter the training to accommodate the partial measurements $\mathbf{y}$ [46, 47], modifying Eq. (1) to

$$\mathcal{L} = \mathbb{E}\left[w_t \left\|m_\theta(\mathbf{x}_t, t, \mathbf{y}) - \mathbf{x}_0\right\|^2\right], \tag{3}$$

where $\mathbf{y} = \boldsymbol{H}\mathbf{x}_0$ for some desired degradation $\boldsymbol{H}$. It is common to implement the conditioning on $\mathbf{y}$ by concatenating $\boldsymbol{H}^\dagger\mathbf{y}$ to the noisy network inputs[2]. $\boldsymbol{H}^T\mathbf{y}$, or a similar operation that shapes the measurements back into the image dimensions can also be used instead, but efficient implementations of both $\boldsymbol{H}$ and $\boldsymbol{H}^\dagger$ exist for many practical degradations. Such methods typically achieve good results, and have proven to be accurate. Yet, such methods are not conditioned on the degradation operator $\boldsymbol{H}$, limiting them to a single or few degradations per trained model [46, 47]. Several works extend these frameworks to flow matching [14] and Schrödinger bridges [34]. Adapting to these methods are beyond the scope of this work, but we expect our architecture to perform equally well for these different training regimes.

An alternative approach is to use an existing unconditional model $m_\theta(\mathbf{x}_t, t)$ and knowledge of $\boldsymbol{H}$ to model the conditional score $\nabla_{\mathbf{x}_t} \log(p(\mathbf{x}_t|\mathbf{y}, \boldsymbol{H}))$, using the Bayes rule

$$\nabla_{\mathbf{x}_t} \log(p(\mathbf{x}_t|\mathbf{y}, \boldsymbol{H})) = \nabla_{\mathbf{x}_t} \log(p(\mathbf{x}_t)) + \nabla_{\mathbf{x}_t} \log(p(\mathbf{y}|\mathbf{x}_t, \boldsymbol{H})). \tag{4}$$

Such methods are referred to as zero-shot [11, 10, 33, 32, 51, 59, 60], for their use of pre-trained diffusion models to solve inverse problem tasks. These methods are highly flexible, for a single trained model can be used to solve any inverse problem for which $\nabla_{\mathbf{x}_t} \log(p(\mathbf{y}|\mathbf{x}_t, \boldsymbol{H}))$ can be approximated. Yet, these methods are often inaccurate, slow, and computationally expensive, due to the challenge of obtaining an accurate approximation for $\nabla_{\mathbf{x}_t} \log(p(\mathbf{y}|\mathbf{x}_t, \boldsymbol{H}))$. In contrast to the correctness guarantees of training-based models [5], zero-shot models cannot sample from the posterior even with an ideal denoiser [22].

## 2.4 Algorithm Unrolling

Algorithm unrolling, also known as deep unfolding, represents a paradigm where iterative algorithms inspired by classical optimization utilize repeated application of neural networks. This approach, first introduced in Gregor and LeCun [21], had emerged into image processing [54, 61, 1, 40]. In inverse problem solving, many algorithm unrolling methods apply $\boldsymbol{H}$ and $\boldsymbol{H}^\dagger$ (or $\boldsymbol{H}^T$) between network evaluations, and are trained in an end-to-end manner. Deep unrolling algorithms are typically trained for distortion reduction and do not accommodate posterior sampling, unlike our method, which is a diffusion model. Also, while the InvFusion architecture is inspired by the same principles, it uses a very different execution. Instead of approximating an iterative algorithm, InvFusion uses the attention mechanism to learn where to utilize the knowledge of the degradation.

## 2.5 Attention for Conditional Generation

Attention mechanisms [57], originally developed for language processing tasks, have emerged as a dominant architectural paradigm in image processing applications. The fundamental operation in attention layers involves matching queries and keys, enabling the network to identify and combine corresponding features across different data streams, and output relevant values. This architectural approach has found widespread adoption in image generation tasks, particularly in conditional generation scenarios. A prominent example is text-to-image synthesis [45, 47], where cross-attention mechanisms facilitate interaction between textual inputs and internal diffusion model activations. The versatility of attention mechanisms extends beyond text-to-image applications, demonstrating significant utility in various domains, such as image editing [25, 56, 9], novel-view-synthesis [55, 18, 48] and many more.

## 3 Method

The integration of degradation operators directly as input to neural networks presents significant computational and architectural challenges. The complexity arises from the vast dimensionality of potential degradation representations. Even when considering only linear degradations, each row of

---

[2]$\boldsymbol{H}^\dagger$ is the Moore–Penrose pseudo-inverse of the operator $\boldsymbol{H}$.

the degradation matrix is as large as the input image itself, creating an overwhelming input space that would render traditional network architectures computationally intractable and practically infeasible.

To address this fundamental challenge, our approach incorporates the degradation operator within the network architecture itself. We introduce a novel *Feature Degradation Layer*, which applies the degradation operator to the internal network activations, and compares the result to the provided measurements, as shown in Fig. 2.

Specifically, in this layer, feature representations undergo the following operations: deep features are first rearranged in the shape of a stack of images, to which the degradation is applied. These degraded features are then concatenated with the measurements, and are transformed with a learned linear operator, incorporating the information from both the measurements and the degraded features.

Finally, the transformed degraded features are mapped back to the feature space through the degradation's pseudo-inverse. This process can be described formally as

$$\tilde{\mathbf{a}} = \boldsymbol{H}^{\dagger}(T_{\psi}([\boldsymbol{H}\mathbf{a} \quad \mathbf{y}])), \qquad (5)$$

where $\mathbf{a} \in \mathbb{R}^{D \times c}$ are the inputs to the layer, and $T_{\psi} : R^{(c+1)} \to R^c$ is a learned transform and activation operating on the channels axis $c$.

We interweave the output of our Feature Degradation Layers back into the network's processing stream using joint-attention[3], as illustrated in Fig. 2. These complete blocks, which we name *InvFusion Blocks*, are designed to replace some or all standard attention blocks in existing Transformer or UNet architectures. The above operations require that the network's deep feature resolution match the input image dimension

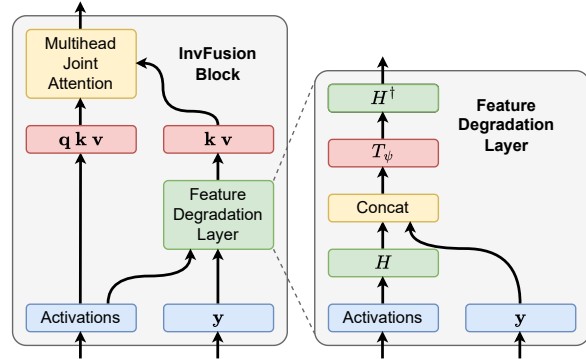

Figure 2: **A diagram of the InvFusion Block.** Our block contains a Feature Degradation Layer, which incorporates the operator $\boldsymbol{H}$ into the architecture by applying $\boldsymbol{H}$ on the activations and comparing them with the measurements $\mathbf{y} = \boldsymbol{H}\mathbf{x}$.

$D$. This constraint is readily accommodated in many Vision Transformer [17, 43, 13] based networks by using a number of hidden channels that is divisible by the original image channels (which does not compromise performance). Internal network activations can then be un-patched back to the original image shapes to apply the degradation $\boldsymbol{H}$.

Finally, using our complete InvFusion architecture, a diffusion model can be trained directly by conditioning on both the degradation and the measurements,

$$\mathcal{L} = \mathbb{E}\left[w_t \left\|m_\theta(\mathbf{x}_t, t, \mathbf{y}, \boldsymbol{H}) - \mathbf{x}_0\right\|^2\right], \qquad (6)$$

where $\boldsymbol{H}$ is sampled from some pre-determined distribution over possible degradations. Additionally, we also find it is beneficial to concatenate $\boldsymbol{H}^{\dagger}\mathbf{y}$ to the noisy network input.

Sampling from InvFusion maintains fundamental compatibility with standard diffusion model sampling procedures, offering a significant advantage in terms of flexibility and implementation. This compatibility means that any established sampling technique developed for conventional diffusion models can be directly applied to InvFusion.

### 3.1 MMSE Estimator

Our InvFusion architecture can be used for more than posterior sampling. Having an architecture that makes use of the degradation $\boldsymbol{H}$ enables training a single model with

$$\mathcal{L} = \mathbb{E}\left[\left\|m_\theta(\mathbf{y}, \boldsymbol{H}) - \mathbf{x}\right\|^2\right] \qquad (7)$$

to produce an accurate MMSE predictor for many different degradations. Alternatively, the MMSE predictor can be approximated by a diffusion model trained with Eq. (6) by applying a single denoising

---

[3]Joint-attention differs from cross-attention by using the same queries for both the "self" and "cross" keys

Table 1: **Comparison of inverse-problem solving methods on FFHQ64.** Best method in bold.

| Method | Strided Motion Blur | | | In-Painting | | | Matrix Operator | | | NFEs |
|---|---|---|---|---|---|---|---|---|---|---|
| | PSNR↑ | FID↓ | CFID↓ | PSNR↑ | FID↓ | CFID↓ | PSNR↑ | FID↓ | CFID↓ | |
| Zero-shot Methods | | | | | | | | | | |
| DDRM [33] | 22.5 | 23.2 | 23.2 | 12.6 | 53.6 | 53.6 | 10.8 | 72.7 | 72.7 | 25 |
| DDNM [59] | 24.0 | 42.5 | 42.3 | 16.2 | 54.6 | 54.6 | 11.9 | 131.4 | 131.4 | 100 |
| DPS [11] | 18.3 | 9.7 | 17.7 | 10.1 | 17.0 | 67.5 | 9.6 | 26.7 | 35.9 | 1000* |
| ΠGDM [51] | 22.1 | 6.3 | 6.3 | 15.2 | 12.8 | 12.8 | 10.7 | 19.1 | 19.1 | 100* |
| DAPS [60] | 17.5 | 143.5 | 60.3 | 16.3 | 16.9 | 19.1 | 12.0 | 56.3 | 87.2 | 1000 |
| MGPS [41] | 20.5 | 7.7 | 9.7 | 13.9 | 10.5 | 24.3 | 11.7 | 8.2 | 29.0 | 1764 |
| Training-based Methods | | | | | | | | | | |
| Palette [46] | 21.0 | 4.7 | 5.8 | 17.0 | 5.3 | 5.5 | 12.5 | **6.6** | 25.5 | 63 |
| InDI [14] | 21.4 | 21.6 | 22.8 | 17.7 | 22.6 | 25.0 | 13.7 | 35.5 | 58.7 | 50 |
| **InvFusion** | 22.7 | **4.2** | **4.2** | 17.1 | **5.0** | **5.0** | 15.1 | 7.1 | **7.1** | 63 |
| MMSE predictor (Training-based) | | | | | | | | | | |
| Palette [46] | 23.5 | - | - | 19.5 | - | - | 15.5 | - | - | 1 |
| **InvFusion** | **25.0** | - | - | **19.8** | - | - | **17.9** | - | - | 1 |

Table 2: **Comparison of inverse-problem solving methods on ImageNet64.** Best method in bold.

| Method | Strided Motion Blur | | | In-Painting | | | Matrix Operator | | | NFEs |
|---|---|---|---|---|---|---|---|---|---|---|
| | PSNR↑ | FID↓ | CFID↓ | PSNR↑ | FID↓ | CFID↓ | PSNR↑ | FID↓ | CFID↓ | |
| Zero-shot Methods | | | | | | | | | | |
| DDRM [33] | 21.2 | 26.1 | 26.2 | 12.2 | 30.5 | 30.5 | 10.6 | 32.9 | 32.9 | 25 |
| DDNM [59] | 22.5 | 69.5 | 69.4 | 15.1 | 95.3 | 95.3 | 11.3 | 115.3 | 115.3 | 100 |
| DPS [11] | 18.3 | 9.0 | 12.9 | 10.4 | 15.2 | 17.7 | 9.5 | 12.8 | 12.8 | 1000* |
| ΠGDM [51] | 20.0 | 7.1 | 7.1 | 14.3 | 8.7 | 8.7 | 10.6 | 8.9 | 8.9 | 100* |
| DAPS [60] | 17.3 | 125.7 | 62.8 | 15.4 | 16.3 | 19.7 | 11.7 | 28.3 | 37.0 | 1000 |
| MGPS [41] | 19.4 | 14.2 | 18.0 | 13.3 | 18.5 | 25.0 | 11.7 | 13.8 | 18.9 | 1764 |
| Training-based Methods | | | | | | | | | | |
| Palette [46] | 19.6 | 6.6 | 7.0 | 15.6 | 6.2 | 6.2 | 12.3 | 6.7 | 11.4 | 63 |
| InDI [14] | 20.6 | 23.3 | 24.6 | 17.1 | 21.0 | 21.0 | 14.5 | 30.5 | 40.0 | 50 |
| **InvFusion** | 20.9 | **5.6** | **5.6** | 15.6 | **5.9** | **5.9** | 14.5 | **6.3** | **6.3** | 63 |
| MMSE predictor (Training-based) | | | | | | | | | | |
| Palette [46] | 22.3 | - | - | 18.3 | - | - | 15.3 | - | - | 1 |
| **InvFusion** | **23.4** | - | - | **18.4** | - | - | **17.3** | - | - | 1 |

* Methods that use the network derivative, using more computation and memory per NFE.

step on white Gaussian noise. That is, for sufficiently small $\bar{\alpha}(t)$ we have that $\mathbf{x}_t \approx \epsilon$, which is independent of $\mathbf{x}_0$, so that

$$\mathbb{E}\left[\mathbf{x}_0 | \mathbf{x}_t, \mathbf{y}, \boldsymbol{H}\right] = \mathbb{E}\left[\mathbf{x}_0 | \mathbf{y}, \boldsymbol{H}\right], \tag{8}$$

which also holds for the approximate models being trained. In other words, any conditional diffusion model can be used as an efficient MMSE predictor. We find that this works well (see App. B), and utilize this in our experiments.

## 4   Experiments

We evaluate the advantage of InvFusion over existing inverse problem solvers on $64 \times 64$ images from the FFHQ [29] and ImageNet [15] datasets. In these experiments, we conduct a comparative analysis of InvFusion against alternative methods that can operate on several types of degradations

with a single model. This includes zero-shot methods, utilizing an comparable unconditional model. Additionally, we evaluate our approach against a "Palette"-style methodology [46] that incorporates the $\boldsymbol{H}^\dagger \mathbf{y}$ as input without additional degradation information. We train these models ourselves using the same underlying architecture, to enable a comprehensive assessment of our method's capabilities disentangled from architectural considerations. The models trained on ImageNet are also class-conditional, and use CFG [27] with a factor of 2.0 for sampling.

We train and evaluate our model on several categories of inverse problems; strided motion blur (combining various degrees of super-resolution and de-blurring), in-painting by leaving only patches of various sizes, and a general matrix degradation operator. These degradations each have computationally efficient implementations for applying both $\boldsymbol{H}$ and $\boldsymbol{H}^\dagger$. The parameters of each degradation are randomized, to create a large corpus of possible degradations (as detailed in App. A). All conditional models are also trained on unconditional generation by using a degradation which always outputs zeros. Figure 3 shows qualitative comparisons of different inverse problem solving methods for the two datasets. All training-based approaches use the deterministic sampler from EDM [30] along with identical seeds, highlighting the effects of different models.

For quantitative comparisons, we use PSNR to measure the image fidelity, *i.e.* how close the restored output is to the original. To measure image realism, we use 10K FID [26]. These two metrics are at odds according to the Perception-Distortion-Tradeoff [8], and different methods excel at either of the metrics. We find that some restoration algorithms seem to sacrifice consistency with the measurements $\mathbf{y}$ to accommodate better image realism, in effect ignoring the inverse problem. To measure the image realism of valid solutions to the given inverse problem, we offer to measure the FID of generated samples that have been projected to be consistent with the measurements $\mathbf{y}$, which we refer to as CFID. In this way, we penalize models for the distance between the generated sample and the nearest valid solution to the formulated problem. We also include LPIPS [62] measures in App. D. Tables 1 and 2 show our qualitative analysis on the FFHQ [29] and ImageNet [15] datasets respectively. Our model achieves the best CFID among all training-based and zero-shot methods, along with the best FID in all but a single case, suggesting it is SOTA in generating samples from the posterior for our setting. This superior quality is attained despite using far less computations than nearly all zero-shot methods, as quantified by Neural Function Evaluations (NFEs). Our InvFusion MMSE estimator based on the same model achieves the best PSNR of all models, while the InvFusion posterior sampler is also among the highest in PSNR.

Interestingly, the gap in performance between InvFusion and Palette highly varies between different degradation families. In problems like in-painting, the degradation is easily inferrable by the network, as the $\boldsymbol{H}^\dagger \mathbf{y}$ input to the network contains masked regions (see Fig. 3). Thus in in-painting, we see a small advantage for InvFusion over Palette. On the other hand, in motion-blur and even more so on general matrix degradations, Palette struggles to infer the exact degradation, leading to a drop in both PSNR and CFID, sometimes, even lower than some zero-shot methods which are degradation-aware. This supports our hypothesis that knowledge of the degradation operator, whether inferred or explicit, is critical for correct restoration.

Previous comparisons focused on smaller images ($64 \times 64$) to enable efficient evaluations of many methods and degradations, some of which are computationally expensive. To demonstrate InvFusion's scalability and performance at higher resolutions, we conducted additional experiments on strided motion blur and patch in-painting using $256 \times 256$ images from FFHQ. The complete results are presented in App. D, with representative examples shown in Fig. 1.

### 4.1 Out-Of-Distribution Degradations and Unconditional Sampling

To measure the generalization capabilities of different methods we evaluate performance on a degradation that is slightly outside of the training distribution. In this case, we perform out-painting from a single rectangular area in the image. The results in Tab. 3, show that Palette completely fails to generalize, despite the similarity to the in-painting task the model has been trained on and the ease at which the degradation operator can be inferred from the input. On the other hand, InvFusion does quite well on this problem, outperforming all zero-shot methods, which are not penalized by the out-of-distribution (OOD) degradation, as they are not trained on any degradation. Notably, the matrix degradations demonstrated in Tabs. 1 and 2 underscore the generalization capabilities of InvFusion, as the range of possible degradations is so huge that it is near certain that none of the degradations used in test-time have appeared in training. Finally, we also evaluate all trained models for unconditional

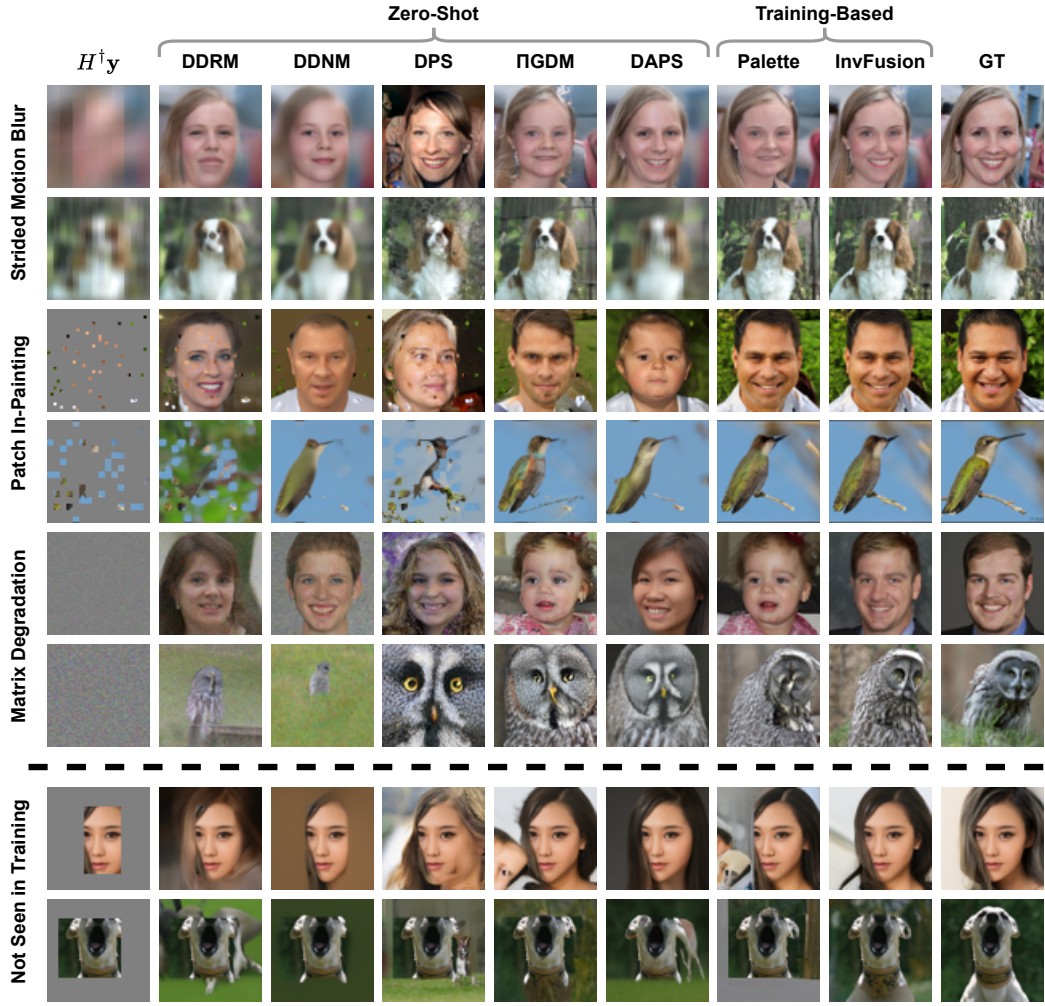

Figure 3: **Examples comparing zero-shot and training-based inverse problem solvers.** For each degradation, the top row is an example from FFHQ64 and the bottom row from ImageNet64. Images generated using training-based methods use deterministic samplers and identical seeds, highlighting the subtle effects the different training algorithms.

generation (on which they had been trained, by using $\boldsymbol{H} \in R^{0 \times D}$). Surprisingly, we find that both conditional models outperform the unconditional baseline, with InvFusion leading with approximately 5% improvement over the baseline (unconditional diffusion). This is despite the conditional models having no additional capacity in unconditional generation, as both the additional model inputs and the Feature Degradation Layer are always output zeros in this setting. We hypothesize that being exposed to partial image information during training helps the diffusion model converge. This indicates that a single InvFusion model can be well adapted for both restoration and unconditional generation.

## 4.2 Noisy Degradations

Following the noiseless linear case explored in earlier sections, we also conduct experiments on noisy inverse problems. We retrain the FFHQ64 model, incorporating additive white Gaussian noise with a standard deviation of $\sigma_n = 0.1$ in the measurements $\mathbf{y}$. The InvFusion architecture remains unchanged, using only the transforms $\boldsymbol{H}$ and $\boldsymbol{H}^\dagger$ of the degradation – as adding randomly sampled noise at each layer could lead to accumulation of noise. Projection operators are not well-defined for noisy degradations, so only PSNR and FID are evaluated for the noisy case. Our findings in Tab. 11 (App. D) demonstrate the advantage of InvFusion over alternative methods in the noisy settings.

Table 3: **Comparison of restoration on a degradation that did not appear in training.** InvFusion demonstrates strong adaptation capabilities through its degradation-aware architecture.

| Method | PSNR↑ | FID↓ | CFID↓ |
|---|---|---|---|
| Zero-shot Methods | | | |
| DDRM [33] | 11.6 | 24.9 | 24.9 |
| DDNM [59] | 12.7 | 75.8 | 75.8 |
| DPS [11] | 10.1 | 8.8 | 14.1 |
| ΠGDM [51] | 11.3 | 7.3 | 7.3 |
| DAPS [60] | 12.1 | 40.8 | 39.5 |
| Training-based Methods | | | |
| Palette [46] | 11.0 | 19.9 | 19.9 |
| **InvFusion** | 11.9 | **5.9** | **5.9** |
| MMSE predictor (Training-based) | | | |
| Palette [46] | 13.6 | - | - |
| **InvFusion** | **14.7** | - | - |

Table 4: **Unconditional sampling FID.**

| | FFHQ | ImageNet |
|---|---|---|
| Baseline [13] | 6.66 | 8.98 |
| Palette [46] | 6.56 | 8.26 |
| **InvFusion** | **6.36** | **8.17** |

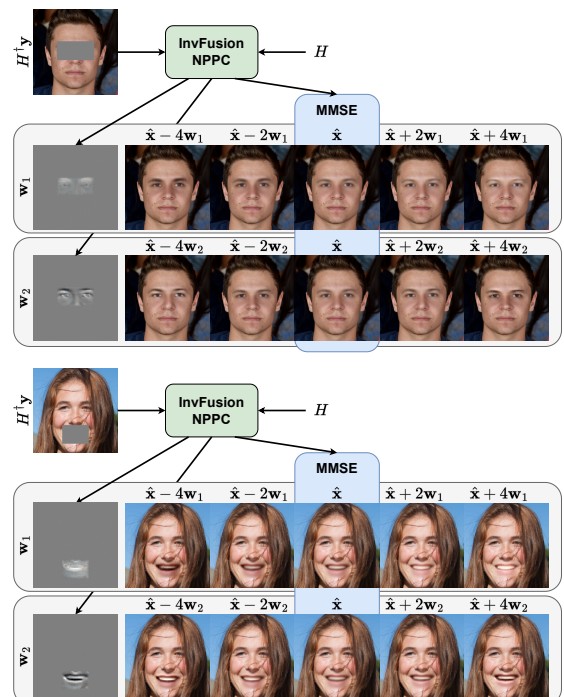

Figure 4: **Using InvFusion NPPC.** By being degradation-aware, the model can be trained to predict the MMSE along with several leading principal components $\mathbf{w}_i$ (left, contrast enhanced) for many degradations.

## 5 Neural Posterior Principal Components

Beyond the improvements in restoration fidelity and realism, using an operator-aware architecture unlocks several capabilities that are useful for down stream applications. A notable example, is the computation of Neural Posterior Principal Components (NPPCs) [42], which are efficient estimations of the posterior distribution's largest principal vectors. These direction are meaningful for uncertainty quantification among many other tasks [38, 20], and may have high implications for medical analysis [6, 19]. In the original work by Nehme et al. [42], a network is trained to approximate the posterior principal components for a single pre-defined degradation. Using InvFusion, we can use the same approach to scale up NPPC training to multiple degradations at once, enabling flexible adaptation to the degradation at runtime. We experiment with applying the exact same NPPC loss from [42], using the InvFusion architecture to condition the network on the degradation operator $H$ for the in-painting of a single area of the image. The examples in Fig. 4 shows that InvFusion meaningfully extends NPPC computation for multiple degradation, making a single model viable for a wide range of problems.

## 6 Discussion, Limitations and Conclusion

Despite achieving state-of-the-art performance in posterior sampling and MMSE estimation, InvFusion faces several important limitations. A primary constraint lies in the model's training scope – although InvFusion demonstrates remarkable adaptability to various degradations, it can only be trained on a finite set of degradation types, beyond which the model may under-perform. Even in the linear case, the space of possible degradations scales quadratically with image dimensions, creating fundamental constraints that would be impractical to address solely through increased model capacity and training time. Additionally, while our current results focus exclusively on linear degradations, we anticipate that InvFusion could potentially generalize to non-linear degradations when appropriately

trained, though this remains to be empirically validated. This in turn would enable the use of InvFusion for latent diffusion, in which non-linear operators apply or approximate the encoder and decoder. A final consideration is the computational and memory intensity of repeated degradation applications, particularly during the training phase. Although this computational overhead may be relatively minor compared to certain zero-shot methods that require additional steps or model derivatives, optimizing these resource requirements represents a key area for future improvement.

This work introduces InvFusion, a novel architectural framework that bridges the long-standing gap between training-based and zero-shot approaches in solving inverse problems with diffusion models. By incorporating the degradation operator directly into the network architecture through the attention mechanism, our method achieves state-of-the-art performance while maintaining the flexibility to handle diverse degradation scenarios. The empirical results demonstrate superior performance on multiple datasets and across various inverse problems, while offering computational efficiency comparable to many existing methods. Beyond its primary application, InvFusion's capability as a general MMSE estimator and its potential for NPPC estimation opens new avenues for downstream applications. While certain limitations remain, this work represents a significant step forward in the field of accurate and efficient inverse problem solving.

## Acknowledgments and Disclosure of Funding

This research was partially supported by the Israel Science Foundation (ISF) under Grants 2318/22, 951/24 and 409/24, and by the Council for Higher Education – Planning and Budgeting Committee.

This work was supported by the Institute of Information & Communications Technology Planning & Evaluation (IITP) grant funded by the Korea government (MSIT) (RS-2025-02304967, AI Star Fellowship(KAIST)), and by the National Research Foundation of Korea under Grant RS-2024-00336454.

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

# Appendices

## A  Implementation Details

### A.1  Architecture and Model Training

Code implementation available at `https://github.com/noamelata/InvFusion`.

Our experiments use data from the FFHQ [29] (CC-BY 4.0 license) and ImageNet [15] datasets. We implement all models using the official implementation of HDiT [13] `ImageTransformerDenoiserModelV2` architecture (MIT license). We use default hyperparameters, with changes listed in Tab. 5. The initial patch size is indicated in the table, with the patch size doubling with the progression along the list shown in the 'Depths' column. The attention type column signified which type of attention was used. 'NAttn$x$' indicates neighborhood attention [23, 24] with a kernel of size $x$, which performs attention between patches only in an $x$ sized neighborhood. 'GAttn' indicates global attention. As described in Sec. 3, InvFusion requires that all layer widths are divisible by the input channels, which in this case is 3. Also each layer's width times the layer's feature map resolution must be a multiple of the input dimension $D$. In our experiments, this held for all but the deepest layers in the FFHQ256 model. For this reason, we did not apply InvFusion to these layers, and used the baseline unconditional attention implementation instead. This provides evidence that our InvFusion architecture works even when not applied evenly on all the network's attention layers. In preliminary experiments, we found joint-attention, in which keys and values from the feature degradation layer are concatenated with the keys and value in the main (self) stream, is preferable to applying self-attention followed by cross-attention.

Models are trained and evaluated with the official implementation of the EDM2 [31] training script (licensed as CC BY-NC-SA 4.0), using the default `P_mean = -0.8, P_std = 1.6`, and a learning rate of $5e-5$, using the default learning rate scheduler. Sampling is done with the default Heun scheduler using a total of 63 NFEs for sampling. Additional experiment specific-training hyperparameters can be found in Tab. 6. The augmentation implementation and hyperparameters are cloned from EDM [30]. We have used 8 Nvidia A40 GPUs (or equivalent hardware) with 49GB of memory for all experiments. Training our model takes approximately one day for the FFHQ64 models, two days for the FFHQ256 model, and four days for the ImageNet64 model.

### A.2  Pseudo-Code

#### A.2.1  Feature Degradation Layer

The Feature Degradation Layer is implemented with the following steps, as described below; The activations are un-patched into the image shape, after which the degradation is applied to them. Next, the measurements are concatenated to the degraded features, a linear function and an activation are applied, and the pseudo-inverse degradation takes the degraded features back into the images space.

Table 5: **Architecture Hyperparameters**

| Experiment | Patch | Depths | Widths | Attention Type |
|---|---|---|---|---|
| FFHQ64 | $(2,2)$ | $[2,2,8]$ | $[384, 768, 1536]$ | $[\text{NAttn7}, \text{NAttn7}, \text{GAttn}]$ |
| ImageNet64 | $(2,2)$ | $[2,2,8]$ | $[384, 768, 1536]$ | $[\text{NAttn7}, \text{NAttn7}, \text{GAttn}]$ |
| FFHQ256 | $(4,4)$ | $[2,2,4,2]$ | $[192, 384, 768, 1536]$ | $[\text{NAttn5}, \text{NAttn7}, \text{GAttn}, \text{GAttn}]$ |

Table 6: **Training Hyperparameters**

| Experiment | Iters | Batch | Aug Prob | Label Dropout | Mixed Precision |
|---|---|---|---|---|---|
| FFHQ64 | $2^{16}$ | 512 | 0.12 | - | fp16 |
| ImageNet64 | $2^{18}$ | 512 | 0.0 | 0.1 | fp16 |
| FFHQ256 | $2^{18}$ | 256 | 0.12 | - | bf16 |

```
class FeatureDegradation(nn.Module):                                      1
    def __init__(self, channels, patch_size, im_channels=3):              2
        super().__init__()                                                3
        self.h = patch_size[0]                                            4
        self.w = patch_size[1]                                            5
        self.im_channels = im_channels                                    6
        self.deg_linear = Linear(channels + 1, channels, bias=True)       7
                                                                          8
    def forward(self, x, degradation, y):                                 9
        x = rearrange(x,                                                  10
            "... h w (nh nw k c) -> ... k c (h nh) (w nw)",               11
            nh=self.h, nw=self.w, c=self.im_channels)                     12
        _y = degradation.H(x)                                             13
        _y = torch.cat([y, _y], -2)                                       14
        _y = act(self.deg_linear(_y))                                     15
        _x = degradation.H_pinv(_y)                                       16
        _x = rearrange(_x,                                                17
            "... k c (h nh) (w nw) -> ... h w (nh nw k c)",               18
            nh=self.h, nw=self.w)                                         19
    return _x                                                             20
```

### A.2.2  InvFusion Block

Below is the implementation of an InvFusion block applied on global attention. The input $\mathbf{x}$ is projected to create queries, values and keys, written as $\mathbf{q}, \mathbf{k}, \mathbf{v}$ respectively. The conditional input $\mathbf{y}$, which is the output of the Feature Degradation layer, is also projected to make additional keys and values, $\mathbf{k'}, \mathbf{v'}$, which are concatenated to the previous keys and values before the attention is computed. We find that projecting both $\mathbf{x}$ and $\mathbf{y}$ to produce $\mathbf{k'}$ is beneficial, as the query key matching may be harder to learn solely on the outputs of the Feature Degradation layers. The neighborhood attention of an InvFusion block is similar, using the native neighborhood attention [23, 24] implementation to compute the attention itself.

```
class SelfAttentionBlock(nn.Module):                                      1
    def __init__(self, d, d_head, cond, dropout=0.0,                      2
            joint=False # Whether to apply InvFusion in this layer        3
        ):                                                                4
        super().__init__()                                                5
        self.d_head = d_head                                              6
        self.n_heads = d // d_head                                        7
        self.norm = AdaRMSNorm(d, cond)                                   8
        self.jnorm = AdaRMSNorm(d, cond) if joint else None              9
        self.qkv_proj = Linear(d, d * 3, bias=False)                     10
        self.jk_proj = Linear(d * 2, d, bias=False) if joint else None   11
        self.jv_proj = Linear(d, d, bias=False) if joint else None       12
        self.scale = nn.Parameter(torch.full([self.n_heads], 10.0))      13
        self.pos_emb = AxialRoPE(d_head // 2, self.n_heads)              14
        self.dropout = nn.Dropout(dropout)                               15
        self.out_proj = zero_init(Linear(d, d, bias=False))             16
                                                                         17
    def forward(self, x, theta, cond, y=None):                           18
        skip = x                                                         19
        x = self.norm(x, cond)                                           20
        qkv = self.qkv_proj(x)                                           21
        q, k, v = rearrange(qkv,                                         22
            "n h w (t nh e) -> t n nh (h w) e",                          23
            t=3, e=self.d_head)                                          24
        q, k = scale(q, k, self.scale[:, None, None], 1e-6)             25
        q = apply_rotary_emb(q, theta)                                   26
        k = apply_rotary_emb(k, theta)                                   27
        if self.jk_proj is not None: # Enter into InvFusion block        28
            y = self.jnorm(y, cond)                                      29
            jk = self.jk_proj(torch.cat([x, y], -1))                    30
            jv = self.jv_proj(y)                                        31
            jk, jv = (rearrange(a,                                       32
```

```
            "n h w (nh e) -> n nh (h w) e",                        33
            e=self.d_head) for a in (jk, jv))                       34
        _, jk = scale(q, jk, self.scale[:, None, None], 1e-6)       35
        jk = apply_rotary_emb_(jk, theta)                           36
        k = torch.cat([k, jk], -2)                                  37
        v = torch.cat([v, jv], -2)                                  38
    x = F.scaled_dot_product_attention(q, k, v, scale=1.0)          39
    x = rearrange(x,                                                40
        "n nh (h w) e -> n h w (nh e)",                             41
        h=skip.shape[-3], w=skip.shape[-2])                         42
    x = self.dropout(x)                                             43
    x = self.out_proj(x)                                            44
    return x + skip                                                 45
```

## A.3 Degradations

Below we explain the implementation details of each family of degradations used in the paper. Several of the degradations include some form of zero-padding to rows of the degradations $\boldsymbol{H}$ (in its matrix form) and therefore to the measurements $\mathbf{y}$ for practical reasons. This is done such that different degradations can be applied in batched form. This produces identical results, as the linear transform $T_\psi$ operates on the channel dimension only, and $\boldsymbol{H}^\dagger$ would similarly have zero-padded columns.

### A.3.1 Patch In-Painting

Patch in-painting is implemented as any in-painting mask where $p \in (0, 0.1)$ of patches of some unvarying size are visible. When creating a new mask, the patch size, $p$, and visible patches are randomly sampled until a valid mask in which some patches are visible is created. We perform this "rejection sampling" to remove masks in which no patches are visible.

The pseudo-inverse of a masking operator is itself, thus the computation of $\boldsymbol{H}^\dagger$ is trivial. In practice, we retain zeros in the masked measurements $\mathbf{y}$, which is equivalent computationally, to enable different masks with different number of masked pixels across the batch dimension.

### A.3.2 Strided Motion-Blur

This family of degradations implements a strided convolution operator, for which we specifically choose a motion-blur kernel. To create a new degradation operator, we first sample a motion-blur kernel the size of the input image using the `motionblur` library.[4] We then sample a stride size from $[2, 4, 8]$ and smooth the motion-blur kernel by convolving it with a isometric Gaussian kernel with standard deviation equal to the stride. The kernels are normalized for numeric stability.

The strided convolution is applied in the frequency space for efficient computation, as this is more efficient for large convolution kernels and simplifies the computation and application of the pseudo-inverse operator. At runtime, the given image or activations are transform to the Fourier domain, where they are multiplied with the pre-computed frequency-space operator.

### A.3.3 Matrix Degradation

The matrix degradation is a matrix operator with $2 - 128$ rows, each of which is sampled from a multivariate normal distribution and normalized. The number of rows is limited to enable the use of different matrices across the batch dimension, which is more computationally intensive. The pseudo-inverse operator is computed directly with pytorch's `p_inv` function.

As any degradation can be represented by a matrix, this family represents all possible degradations in theory. Yet, we find that a model trained only on randomly sampled matrices as described here does not generalize well to specific tasks such as in-painting, super-resolution, or de-blurring.

### A.3.4 Box Out-Painting

This degradation is similar to the patch in-painting, but instead of masking all but several patches the box out-painting leaves only a single rectangle of the image visible. To create this degradation we

---

[4]https://github.com/LeviBorodenko/motionblur

Table 7: **Comparison of MMSE training and diffusion models used for MMSE.**

| Architecture & Loss | Strided Motion Blur PSNR↑ | In-Painting PSNR↑ | Matrix Operator PSNR↑ |
|---|---|---|---|
| Trained for MMSE (Eq. (7)) | | | |
| Palette [46] | 23.08 | 19.36 | 15.63 |
| InvFusion | 24.66 | 19.79 | 17.92 |
| Trained For diffusion (Eq. (6)) | | | |
| Palette [46] | 23.47 | 19.65 | 15.51 |
| InvFusion | 24.83 | 19.79 | 17.60 |

uniformly sample two different coordinates in the image and mask all pixels outside the rectangle that is formed. The practical implementation details are identical to the patch in-painting degradation.

## A.4 Zero-Shot Methods

We use our own implementations for all zero-shot methods, using the default hyperparameters unless specified otherwise. We use the variance-exploding notation, and we adjust all sampling algorithms accordingly. DPS [11] and DAPS [60] only require access to the degradation operator $\boldsymbol{H}$, while DDNM [59] and ΠGDM [51] also make use of the pseudo-inverse operator $\boldsymbol{H}^{\dagger}$. In the original implementation DDRM [33] makes use of the complete SVD of the operator $\boldsymbol{H}$. Because obtaining this SVD is computationally expensive (or even practically impossible in reasonable time) for most of our degradations, we instead implement the DDRM algorithm using only $\boldsymbol{H}$ and $\boldsymbol{H}^{\dagger}$. For the noise-less case, this implementation is equivalent to the original implementation. This is not true in the general noisy case, unless all the degradation's singular values that are not zeros are equal. Due to the degradation families we choose to apply, we assume that most of the singular values that are not zeros are nearly one for the purpose of implementing DDRM in the noisy degradation experiment. For ΠGDM, we make a similar assumption to implement the noisy sampling algorithm, to avoid inverting a large matrix. For the in-painting case, the assumption holds. Similarly, in the strided motion-blur we make use of the fact that white Gaussian noise remains white Gaussian noise under a Fourier transform, and that the application of the kernel is multiplicative.

## B Using Diffusion Models for MMSE

In Sec. 3.1 we explore the use of the InvFusion architecture of MMSE estimation. A network can be trained directly as a degradation-aware MMSE estimator using a regression loss as seen in Eq. (7). Nevertheless, we notice that we can utilize our existing trained diffusion models (trained using Eq. (6)) for the same task. Intuitively, this is because for sufficiently high noise values, the noisy input to the network is equivalent to pure noise. Therefore, The network learns to rely only on the information from $\mathbf{y}$ and $\boldsymbol{H}$. In practice, we use the value $\sigma_t = 100$ for the MMSE estimation.

We test the gap between models trained directly for MMSE estimation (Eq. (7)) and models trained for diffusion (Eq. (6)) on FFHQ64, using the same architecture. The MMSE model's input is Gaussian noise, instead of the noisy image input to the diffusion model, to keep the architecture identical. The results in Tab. 7 reflect that both methods yield approximately the same results, whether trained using the degradation-aware InvFusion architecture and loss or without it. We conclude that it is probably more cost-effective to use a single model for both posterior sampling and MMSE estimation task, and do so for the experiments in our paper.

## C Ablations

Table 8 shows the effect of applying the InvFusion block only on a fraction of network layers. In this experiment, the InvFusion block has been added or removed selectively to each of the HDiT layers in a specified resolution as used in our base architecture. The results show a gradual improvement across metrics with added applications of the InvFusion block.

Table 8: **Ablation of removing the InvFusion block from the architecture at different resolutions.**

| InvFusion Resolutions | Strided Motion Blur | | | In-Painting | | | Matrix Operator | | |
|---|---|---|---|---|---|---|---|---|---|
| | PSNR↑ | FID↓ | CFID↓ | PSNR↑ | FID↓ | CFID↓ | PSNR↑ | FID↓ | CFID↓ |
| None (Baseline) | 21.0 | 4.7 | 5.8 | 17.0 | 5.3 | 5.5 | 12.5 | 6.6 | 25.5 |
| Resolution 1 | 22.3 | 4.5 | 4.5 | 17.1 | 4.9 | 4.9 | 14.6 | 8.9 | 8.9 |
| Resolutions 1 & 2 | 22.7 | 4.4 | 4.4 | 17.1 | 5.0 | 5.0 | 15.0 | 7.9 | 7.9 |
| Resolutions 1 - 3 (All) | 22.7 | 4.2 | 4.2 | 17.1 | 5.0 | 5.0 | 15.1 | 7.1 | 7.1 |

Table 9: **LPIPS evaluation**

| Dataset | Strided Motion Blur | | In-Painting | | Matrix Operator | |
|---|---|---|---|---|---|---|
| | FFHQ | ImageNet | FFHQ | ImageNet | FFHQ | ImageNet |
| Zero-shot Methods | | | | | | |
| DDRM [33] | 0.095 | 0.246 | 0.268 | 0.458 | 0.517 | 0.277 |
| DDNM [59] | 0.105 | 0.373 | 0.171 | 0.406 | 0.672 | 0.300 |
| DPS [11] | 0.136 | 0.233 | 0.276 | 0.462 | 0.480 | 0.281 |
| PiGDM [51] | 0.064 | 0.156 | 0.158 | 0.312 | 0.470 | 0.287 |
| DAPS [60] | 0.683 | 0.736 | 0.148 | 0.327 | 0.535 | 0.263 |
| MGPS [41] | 0.250 | 0.441 | 0.193 | 0.371 | 0.085 | 0.193 |
| Training-based Methods | | | | | | |
| Palette [46] | 0.068 | 0.147 | 0.103 | 0.215 | 0.409 | 0.236 |
| InvFusion | **0.053** | **0.122** | **0.101** | **0.213** | **0.340** | **0.182** |
| MMSE predictor (Training-based) | | | | | | |
| Palette [46] | 0.126 | 0.327 | 0.144 | 0.318 | 0.606 | 0.314 |
| InvFusion | 0.106 | 0.278 | 0.143 | 0.315 | 0.469 | 0.237 |

# D    Additional Results

## D.1    LPIPS Evaluation

We include LPIPS [62] evaluations for our experiments in Tab. 9, which correspond to the results shown in Tabs. 1 and 2 in Sec. 4. InvFusion demonstrated superior performance on LPIPS as well.

## D.2    Consistency Comparison

To validate the consistency of different methods directly, we measure the MSE in the degradation range-space for each reconstruction method. The results shown in Tab. 10 demonstrate the consistency of the reconstructions corresponding to the experiment in Tab. 2. MSE is used to accommodate degradations of differing rank and range, and therefore has a different scale for each degradation class. The MSE values confirm that InvFusion achieves consistency comparable to methods that employ explicit projections, such as DDNM or DDRM. These results offer a complementary metric to the FID/CFID analysis.

## D.3    Noisy Degradations

Table 11 shows the results of the experiment described in Sec. 4.2. The results show that InvFusion maintains superiority when noise is added to the degradation. We expect this framework to work well across many noise types, not limited to white Gaussian noise, as the model learns to treat the noise through training, which does not rely on any fundamental properties of the noise. Nevertheless, for high noise levels InvFusion fails, as seen in the case for high levels of Poisson noise shown in Tab. 12.

Table 10: **Consistency comparison using MSE in the degradation space.**

| Method | Strided Motion Blur | In-Painting | Matrix Operator |
|---|---|---|---|
| DDRM [33] | 6.90 | 0.00 | 0.00 |
| DDNM [59] | 2.28 | 0.00 | 0.00 |
| DPS [11] | 789.63 | 10.27 | 0.00 |
| ΠGDM [51] | 3.22 | 0.00 | 0.00 |
| DAPS [60] | 720765.61 | 2.89 | 5.68 |
| MGPS [41] | 2087.28 | 23.55 | 9.30 |
| Palette [46] | 115.67 | 0.23 | 11.57 |
| InvFusion | 5.94 | 0.01 | 0.00 |

Table 11: **Comparison of noisy inverse-problem solving methods on FFHQ64.**

| Method | Strided Motion Blur | | In-Painting | | Matrix Operator | | NFEs |
|---|---|---|---|---|---|---|---|
| | PSNR↑ | FID↓ | PSNR↑ | FID↓ | PSNR↑ | FID↓ | |
| Zero-shot Methods | | | | | | | |
| DDRM [33] | 11.7 | 137.0 | 12.2 | 155.6 | 11.8 | 87.0 | 25 |
| DDNM [59] | 19.8 | 97.2 | 17.3 | 46.4 | 11.9 | 136.9 | 100 |
| DPS [11] | 7.0 | 12.3 | 8.3 | 7.8 | 9.2 | 22.9 | 1000[*] |
| ΠGDM [51] | 16.5 | 11.5 | 15.7 | 21.5 | 10.8 | 20.6 | 100[*] |
| DAPS [60] | 16.9 | 46.4 | 15.5 | 43.7 | 10.6 | 76.8 | 1000 |
| Training-based Methods | | | | | | | |
| Palette [46] | 17.1 | 6.0 | 16.7 | 5.2 | 12.6 | 6.6 | 63 |
| **InvFusion** | 21.7 | **4.3** | 16.6 | **5.0** | 14.7 | **6.2** | 63 |
| MMSE predictor (Training-based) | | | | | | | |
| Palette [46] | 19.8 | - | 18.9 | - | 15.6 | - | 1 |
| **InvFusion** | **24.1** | - | **19.4** | - | **17.5** | - | 1 |

[*] Methods that use the network derivative, using more computation and memory per NFE.

## D.4 FFHQ 256

Table 13 shows quantitative results for our experiment on $256 \times 256$ FFHQ images. InvFusion remains the SOTA method as measured by CFID – the perceptual image quality of valid solutions to the inverse problem.

## D.5 Comparison to Algorithm Unrolling

In this experiment, we compare our method to algorithm unrolling [21, 54, 61, 1, 40]. Using the unconditional model trained for the experiments in Sec. 4, we unroll the DDRM [33] algorithm through 10 steps and train this in an end-to-end fashion with MSE loss. Table 14 shows the results obtained by DDRM before and after the finetuning, compared to InvFusion. The results show that InvFusion is far superior in terms of image quality, and can reach higher fidelity using a single NFE using the MSE estimator method.

Table 12: **Comparison of inverse-problem with high level of Poisson noise on FFHQ64.**

| Method | Strided Motion Blur | | In-Painting | |
|---|---|---|---|---|
| | PSNR↑ | FID↓ | PSNR↑ | FID↓ |
| Palette [46] | 14.1 | 17.6 | 12.6 | 9.9 |
| InvFusion | 11.9 | 60.0 | 14.7 | 11.5 |

Table 13: **Comparison of inverse-problem solving methods on FFHQ256.**

| Method | Strided Motion Blur | | | In-Painting | | | NFEs |
| | PSNR↑ | FID↓ | CFID↓ | PSNR↑ | FID↓ | CFID↓ | |
|---|---|---|---|---|---|---|---|
| Zero-shot Methods | | | | | | | |
| DDRM [33] | 25.3 | 43.0 | 43.0 | 12.8 | 101.6 | 101.6 | 25 |
| DDNM [59] | 26.2 | 49.9 | 49.9 | 15.2 | 102.1 | 102.1 | 100 |
| DPS [11] | 22.6 | 15.2 | 19.7 | 9.8 | 24.3 | 90.7 | 1000[*] |
| ΠGDM [51] | 24.6 | 9.4 | 9.4 | 16.6 | 26.4 | 26.4 | 100[*] |
| DAPS [60] | 9.4 | 397.8 | 145.7 | 17.8 | 40.1 | 46.8 | 1000 |
| MPGS [41] | 26.0 | 26.5 | 25.9 | 18.2 | 14.8 | 15.2 | 1764 |
| Training-based Methods | | | | | | | |
| Palette [46] | 23.2 | 8.9 | 12.1 | 18.2 | **11.1** | 12.4 | 63 |
| InDI [14] | 24.1 | 11.3 | 14.2 | 20.6 | 14.0 | 14.1 | 50 |
| InvFusion | 24.4 | **8.4** | **8.4** | 18.5 | 11.2 | **11.2** | 63 |
| MMSE predictor (Training-based) | | | | | | | |
| Palette [46] | 25.4 | - | - | 20.7 | - | - | 1 |
| **InvFusion** | **26.4** | - | - | **21.1** | - | - | 1 |

[*] Methods that use the network derivative, using more computation and memory per NFE.

Table 14: **Comparison of our method with algorithm unrolling.**

| Method | Strided Motion Blur | | | In-Painting | | | Matrix Operator | | | NFEs |
| | PSNR↑ | FID↓ | CFID↓ | PSNR↑ | FID↓ | CFID↓ | PSNR↑ | FID↓ | CFID↓ | |
|---|---|---|---|---|---|---|---|---|---|---|
| DDRM (Sampling) | 22.2 | 29.5 | 29.5 | 12.2 | 92.0 | 92.0 | 11.0 | 91.4 | 91.4 | 10 |
| Unrolled DDRM | 24.1 | 41.2 | 40.0 | 19.6 | 65.6 | 65.6 | 14.1 | 182.9 | 182.9 | 10 |
| InvFusion (Sampling) | 22.7 | 4.2 | 4.2 | 17.1 | 5.0 | 5.0 | 15.1 | 7.1 | 7.1 | 63 |
| InvFusion-MSE | 25.0 | - | - | 19.8 | - | - | 17.9 | - | - | 1 |

## D.6 Comparison to Deep Image Prior

In this section, we compare our method to a recent Deep Image Prior (DIP) method, ASeq-DIP [3]. Methods like Deep Image Prior operate at the single-image level, making restoration a computationally prohibitive task for more than a few images. Additionally, these methods typically optimize for different objectives—such as MAP estimation rather than posterior sampling quality—and employ fundamentally different architectures, limiting the conclusiveness of direct comparisons. To provide meaningful context nonetheless, we evaluated PSNR, KID [7] and LPIPS [62] metrics for 1024 images restored using ASeq-DIP [3], and compare it to our methods in Tab. 15. The results demonstrate a clear advantage for our model, despite the substantially cheaper evaluation. We use the default hyperparameters as described in the official implementation, and we find that using a larger model or a different number of inner and outer steps has a negligible effect on the results.

## D.7 Unconditional Samples

Figures 5 and 6 show qualitative examples for unconditional samples from our model *i.e.* samples that are not conditioned on a measurement vector $\mathbf{y}$. The models trained on ImageNet are conditioned on the class label, and use CFG [27] coefficient of 2.0 to enhance the image quality.

Table 15: **Comparison of our method with Deep Image Prior.**

| Method | Strided Motion Blur | | | In-Painting | | | Matrix Operator | | |
| --- | --- | --- | --- | --- | --- | --- | --- | --- | --- |
| | PSNR↑ | KID↓ | LPIPS↓ | PSNR↑ | KID↓ | LPIPS↓ | PSNR↑ | KID↓ | LPIPS↓ |
| ASeq-DIP [3] | 22.4 | 51.2 | 0.48 | 13.1 | 75.5 | 0.66 | 12.7 | 247.0 | 0.93 |
| InvFusion | 21.1 | 6.3 | 0.13 | 15.5 | 7.6 | 0.24 | 15.2 | 10.0 | 0.34 |
| InvFusion-MSE | 23.7 | 25.71 | 0.33 | 18.44 | 37.9 | 0.39 | 18.1 | 43.8 | 0.50 |

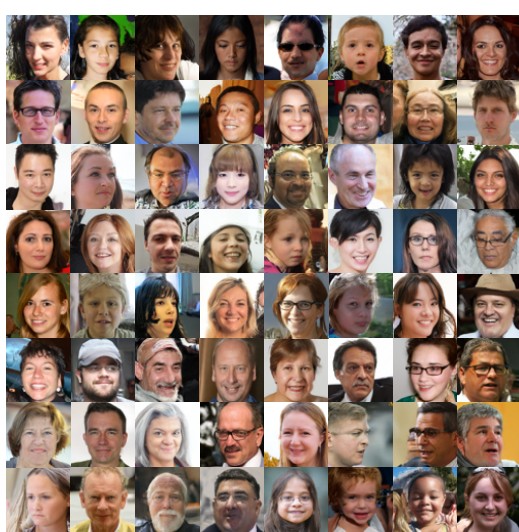

Figure 5: **Examples of unconditional samples from FFHQ generated using our model.**

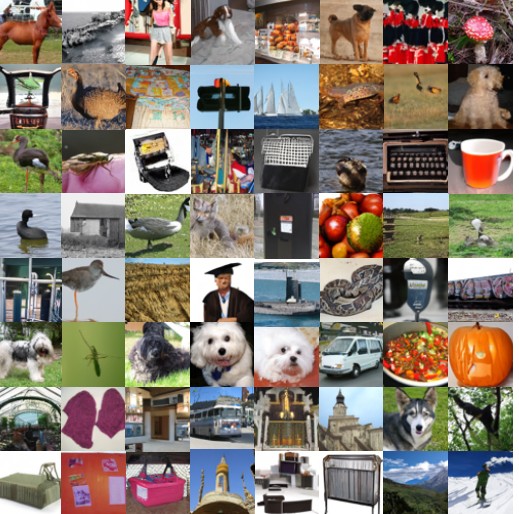

Figure 6: **Examples of class-conditional samples from ImageNet generated using our model.** Samples are not conditioned on any measurements **y**, and are sampled with CFG of 2.0 on the class conditioning.

