# OpenReview forum: "InvFusion: Bridging Supervised and Zero-shot Diffusion for Inverse Problems"
_NeurIPS.cc/2025/Conference — NeurIPS 2025 poster_

### Official Review · Reviewer_21uD · 2025-06-26

**Clarity:** 3
**Significance:** 2
**Originality:** 3
**Rating:** 3
**Confidence:** 3

**Summary:**

This work presents InvFussion, a framework integrating degradation operators into diffusion models via attention layer unrolling to solve inverse problems (e.g., motion blur, in-painting) uniformly. Experiments on FFHQ and ImageNet show InvFussion outperforms zero-shot methods and "Palette" approaches, demonstrating superior performance. Furthermore, InvFussion can efficiently handle both posterior sampling and MMSE estimation, highlighting its computational efficiency and versatility.

**Questions:**

Including additional experiments on generalization to unseen degradation operators would greatly enhance the significance of this work and positively impact my overall assessment. If any of the above points reflect misunderstandings, please clarify them.

**Ethical Concerns:**

["NO or VERY MINOR ethics concerns only"]

**Final Justification:**

The authors have addressed my concerns about the module’s effectiveness through additional experiments in the rebuttal. However, my main concern remains the significance of the method. If the goal is optimal performance, task-specific approaches may be more suitable. If generalization is the aim, the method seems limited to a narrow family of degradations. I encourage the authors to further investigate which types of degradations the method can generalize to, as this is key to evaluating its broader applicability. Overall, thank you for the rebuttal, and I decide to keep my score unchanged.

**Limitations:**

The authors provide a discussion of the method’s limitations in the main text.

**Paper Formatting Concerns:**

No.

**Quality:**

2

**Strengths And Weaknesses:**

**Strengths:**

Originality:
This work presents a novel idea by integrating degradation operators via attention layers, offering a novel perspective for designing flexible and high-performing image generation algorithms. Extensive experiments support the effectiveness of the proposed module.

Clarity:
The manuscript is clearly written, with a well-organized structure and informative figures. The experimental settings are described in sufficient detail, making the work highly reproducible.

**Weaknesses:**

Significance:
While the motivation is solid, the proposed method falls short of its stated goals. It requires training on a specific range of degradation operators and is only generalized to an outpainting task, which closely resembles patch inpainting. In contrast, task-specific methods offer high performance, while zero-shot approaches provide greater flexibility, lower training costs, and no need for paired data.
The proposed method limits its practical applications compared to these two paradigms.

Quality:
1. Comparison experiments lack adequacy as the proposed method is only compared with a training-based approach published three years ago.
2. The outpainting task used to demonstrate generalization shares notable similarity with the training task. It would be more convincing to test cross-domain generalization—for example, training on inpainting and evaluating on super-resolution or deblurring.
3. The paper lacks ablation studies, which would help validate the method’s effectiveness. For example, removing the forward operator $H$ and only inputting $H^\top y$ into $T_\psi$ could clarify the contribution of incorporating the degradation operator.

---

> ### Author Rebuttal · Authors · 2025-07-29
>
> Thank you for your comprehensive review and constructive feedback. We address each concern below:
>
> * **Generalization Capabilities** We acknowledge this limitation and appreciate the opportunity to clarify our contribution's scope. The reviewer correctly notes that our method operates within a trained distribution of degradations, which differs from the unlimited flexibility of zero-shot methods. However, our approach occupies a valuable middle ground with specific practical advantages. \
> Within the trained degradation family, our method demonstrates strong generalization - for instance, in our random matrix experiments reported in Tables 1 and 2, the degradation space is so vast that test-time degradations are virtually guaranteed to be unseen during training, yet our method significantly outperforms alternatives. Additionally, Section 4.1 demonstrates generalization to tasks slightly outside the training distribution (outpainting, which shares structural similarities with inpainting but involves different spatial patterns). \
> The practical value lies in scenarios where: (1) a known family of degradations is encountered frequently enough to justify training, and (2) computational efficiency and quality gains are important for deployment. This represents a meaningful trade-off between task-specific methods and zero-shot approaches.
>
> * **Alternative Approaches** We acknowledge that our current training-based comparisons are limited. Following the suggestion of reviewers 8Xy4 we include a comparison to InDI [1], a flow-based restoration method, in our multi-degradation setup. Preliminary results corresponding to Table 1 can be found below.
>
> | Method | Deblur | | | In-Painting | | | Matrix | | | NFEs |
> |-|-|-|-|-|-|-|-|-|-|-|
> |                | PSNR    | FID    | CFID    | PSNR    | FID    | CFID    | PSNR    | FID    | CFID    | |
> |InDI [1]            | 21.4    | 21.6    | 22.8    | 17.7    | 22.6    | 25.0    | 13.7    | 35.5    | 58.7    | 50   |
> |InvFussion        | 22.7  | 4.2   | 4.2   | 17.1  | 5.0   | 5.0   | 15.1  | 7.1   | 7.1   | 63     |
>
> * **Ablations** We will include comprehensive ablation studies in the appendix. Specifically, we address the suggested experiment of removing the forward operator $\boldsymbol{H}$ and providing only $\mathbf{y}$ as input to $T_\psi$, which we have implemented for the FFHQ64 experiment and include below. This will help isolate the contribution of our mechanism.
>
> |Method            | Deblur    ||| In-Painting        ||| Matrix |||
> |-|-|-|-|-|-|-|-|-|-|
> |                | PSNR    | FID    | CFID    | PSNR    | FID    | CFID    | PSNR    | FID    | CFID    |
> |Only $\mathbf{y}$     | 21.2    | 4.6    | 5.5    | 17.0    | 5.1    | 5.2    | 12.7    | 6.1    | 24.0    |
> |InvFussion    | 22.7  | 4.2   | 4.2   | 17.1  | 5.0   | 5.0   | 15.1  | 7.1   | 7.1   |
>
> [1] Delbracio et al., "Inversion by direct iteration: An alternative to denoising diffusion for image restoration." arXiv preprint arXiv:2303.11435 (2023).

---

> > ### Comment · Reviewer_21uD · 2025-08-06
> >
> > Thank you for the authors' rebuttal, which addresses some of the earlier concerns. However, my main reservations regarding the method’s generalization and practical relevance remain. As acknowledged by the authors, the method has limited generalization and mainly works within the trained degradation distribution, with slight extensions to similar tasks like outpainting. This scalability lacks theoretical support, making it difficult to define other meaningful degradation families in real-world scenarios. Moreover, a unified approach that handles deblurring, inpainting, and outpainting simultaneously seems to lack clear practical use cases. As a result, the method appears less compelling than zero-shot approaches with stronger generalization or task-specific methods that can achieve higher performance.

---

> > > ### Author Response · Authors · 2025-08-06
> > >
> > > We thank the reviewer for their continued engagement and reconsideration of our method.
> > >
> > > We acknowledge the benefits of both zero-shot and task-specific methods. Nevertheless, we respectfully maintain that our approach is well-suited for our framework, as degradation-aware training provides benefits even within specific degradation families.
> > >
> > > As highlighted in our original manuscript, our method unlocks downstream applications like uncertainty quantification with NPPC, which are impossible to replicate with zero-shot methods alone. The alternative task-specific approach may prove too inflexible. A task-specific model would require retraining for each possible $\boldsymbol{H}$, or suffer from reduced accuracy, even when $\boldsymbol{H}$ belongs to a single degradation class (e.g., different blur kernels within deblurring). We hope the reviewer will consider the merit of our middle path as a novel paradigm that advances the field.

---

### Official Review · Reviewer_XFfT · 2025-07-02

**Clarity:** 4
**Significance:** 4
**Originality:** 4
**Rating:** 5
**Confidence:** 5

**Summary:**

The paper addresses the problem of solving inverse problems using denoising diffusion models. Existing zero-shot approaches can approximate samples from the posterior distribution without requiring operator-specific training, but they often rely on crude approximations that make them slow and potentially inaccurate. On the other hand, training diffusion models for conditional generation yields more accurate results but is computationally expensive and tied to a specific forward operator. The proposed method strikes a balance between these two extremes by introducing a small, trainable component into a pre-trained diffusion model. This component leverages the closed-form expression of the forward operator, enabling more efficient inference while retaining some robustness to moderate distributional shifts in the operator.

**Questions:**

see below

**Ethical Concerns:**

["NO or VERY MINOR ethics concerns only"]

**Final Justification:**

The authors have clarified most of my concerns. The paper is interesting and I have no doubt that it will have a good impact. I recommend it for acceptance.

**Limitations:**

I have a few questions regarding the method:

- The InvFussion block requires the pseudo-inverse of the operator. How would you handle non-linear inverse problems for which the pseudo-inverse is intractable?

- It seems that currently this method does not apply to latent diffusion. I see two bottlenecks for doing so. The operator would be the composition of the operator with the decoder and it is not clear how you would evaluate the activations at the decoder. Next, during training, you would need to backpropagate through the input decoder which is very expensive. Am i correct in assuming that these are the main limitations?

- The noisy observation setting is used with a small Gaussian noise. I assume that the method struggles with high noise levels? How would it behave with different types of noise such as Poisson?

**Paper Formatting Concerns:**

No concerns

**Quality:**

4

**Strengths And Weaknesses:**

Significance and originality: The idea proposed in this paper is original, meaningful, and likely to inspire further developments in the area of inverse problems with diffusion models.

Empirical results: The experimental evaluation is, in my view, quite thorough. The proposed method is compared against several zero-shot diffusion posterior sampling baselines as well as Palette. It consistently outperforms these methods, with the improvement over Palette being more modest. However, the authors convincingly demonstrate that their approach is more robust to out-of-distribution (OOD) operators. A notable weakness of the evaluation is the absence of comparisons with more recent and higher-performing posterior samplers, such as MMPS [1] and MGPS [2]. The baselines considered here are either unstable or known to underperform on tasks like inpainting.

It seems to that there are some weaknesses/limitations for the method but since I am still not 100% sure about them, I list them in the limitations section below.

[1] Learning diffusion priors from observations by expectation maximization. Rozet, F., Andry, G., Lanusse, F. and Louppe, G., 2024. Learning diffusion priors from observations by expectation maximization.

[2] Moufad, B., Janati, Y., Bedin, L., Durmus, A., Douc, R., Moulines, E. and Olsson, J., 2024. Variational Diffusion Posterior Sampling with Midpoint Guidance.

---

> ### Author Rebuttal · Authors · 2025-07-29
>
> Thank you for your comprehensive review and constructive feedback. We address each concern below:
>
> * **Additional Method** We acknowledge that adding more comparisons to these recent posterior samplers would strengthen our paper. Following the reviewer’s suggestion, we have evaluated MGPS [2] on our experimental setting. The results, corresponding to Tables 1 and 2 in our paper, are presented below:
>
> | Method | Deblur | | | In-Painting | | | Matrix | | | NFEs |
> |---------|---------------------|---|---|-------------------|---|---|-------------------|---|---|------|
> |         | PSNR | FID | CFID | PSNR | FID | CFID | PSNR | FID | CFID |      |
> | **FFHQ** |||||||||| |
> | MGPS    | 20.5 | 7.7 | 9.7  | 13.9 | 10.5| 24.3 | 11.7 | 8.2 | 29.0 | 1764 |
> | InvFussion | 22.7  | 4.2   | 4.2   | 17.1  | 5.0   | 5.0   | 15.1  | 7.1   | 7.1   | 63     |
> | **ImageNet** |||||||||| |
> | MGPS | 19.4 | 14.2| 18.0 | 13.3 | 18.5| 25.0 | 11.7 | 13.8| 18.9 | 1764 |
> | InvFussion | 20.9 | 5.6 | 5.6 | 15.6 | 5.9 | 5.9 | 14.5 | 6.3 | 6.3 | 63 |
>
> * **Non-linear Operators** The reviewer raises an important point about pseudo-inverse tractability. Due to the exposure in training, we believe the model would be able to learn to use the operator even with a roughly approximated pseudo-inverse. The joint-attention scheme can also be applied directly with the transformed degraded measurements, without casting them back to the image space (that is, skipping the application of $\boldsymbol{H}$). We hope to explore many non-linear operators in future work.
>
> * **Latent Diffusion** The reviewer correctly identifies the main computational bottlenecks for extending our approach to latent diffusion models. We will clarify these limitations in our revised manuscript. However, recent work such as [3] demonstrates promising approaches for handling operators in latent space that could potentially be adapted to our framework. We note that latent diffusion essentially introduces a non-linear encoding/decoding operator that could be handled using similar principles to those outlined above.
>
> * **Noise Level** We have not observed that the method is limited to certain levels or types of noise, we have simply used a single moderate value of $\sigma$ in our experiment. We expect InvFussion to be more beneficial as the noise level increases, as the blind restoration problem becomes harder. We will add an experiment with a high rate of Poisson noise to the final version to assess our hypothesis.
>
> [3] Raphaeli, Ron, Sean Man, and Michael Elad. "Silo: Solving inverse problems with latent operators." arXiv preprint arXiv:2501.11746 (2025).

---

> > ### Comment · Reviewer_XFfT · 2025-08-05
> >
> > Thank you for the clarifications and additional experiments. I will recommend acceptance of the paper.

---

### Official Review · Reviewer_8Xy4 · 2025-07-03

**Clarity:** 2
**Significance:** 3
**Originality:** 3
**Rating:** 4
**Confidence:** 4

**Summary:**

This paper proposes an architectural innovation for diffusion-based inverse problem solving, where a representation of the degradation operator is incorporated into the architecture. As a result, after training, the model can leverage degradation information in inference time, which may benefit model performance against techniques where the degradation needs to be inferred from the measurement itself. Numerical experiments demonstrate that the proposed technique outperforms zero-shot baselines and a supervised baseline on 2 image datasets with 64$\times$64 resolution.

**Questions:**

- How does the method compare to other techniques on larger resolution images?
- How does performance compare to an end-to-end trained non-diffusion image restoration method with the same training data, such as Restormer? I believe this would provide a strong baseline with highly reduced compute cost compared to diffusion models.
- How does performance compare in terms of LPIPS?
- How does performance scale with respect to NFEs?
- How does compute cost compare across different methods?

**Ethical Concerns:**

["NO or VERY MINOR ethics concerns only"]

**Final Justification:**

Authors added experiments comparing to relevant work and presented an additional metric for comparison. I raised my score in accordance with these improvements.

**Limitations:**

Limitations are discussed in the paper.

**Quality:**

3

**Strengths And Weaknesses:**

Strengths:
- The idea of incorporating unrolled networks into diffusion architectures to better capture the degradation operator is interesting. It is a promising way of conditioning on the degradation.
- The experimental results show clear benefits over zero-shot baselines in terms of performance.
- The paper is easy to follow.

Weaknesses:
- The experiments are performed on low resolution images. It is unclear how the proposed technique performs and compares to others on more practical resolutions such as 256$\times$256.
- Experiments are missing on compute cost and analysis of performance with respect to NFEs.
- There are multiple other learning-based diffusion solvers that have not been compared to in the paper, including [1], [2], [3].
- I don't see how the proposed method is more "degradation-aware" than other techniques that condition on the pseudo-inverse reconstruction of the measurement. On a practical level, I understand that due to the architectural changes, the conditioning may be more efficient/stronger, but sampling is still not conditioned on the operator itself, but only on its action on the input and internal representations. Therefore, on a conceptual level this is not more degradation-aware than other techniques leveraging the action of the operator. A truly degradation-aware method would incorporate some direct representation of the operator explicitly in the prediction. I understand that this may not be practical, but I push back on the claim in the abstract that this is "the first training-based degradation-aware posterior sampler".
- It would be interesting to add LPIPS as an evaluation metric, which measures perceptual quality but is not sensitive to the number of samples.
- Minor comments: (1) I would use the term cross-attention instead of joint-attention. (2) Why is the method called InvFussion and not InvFusion? Typos on lines 63, 121, 130, caption of Fig 4.

[1] Fabian et al.,  "DiracDiffusion: Denoising and incremental reconstruction with assured data-consistency." Proceedings of machine learning research 235 (2024): 12754.

[2] Delbracio et al., "Inversion by direct iteration: An alternative to denoising diffusion for image restoration." arXiv preprint arXiv:2303.11435 (2023).

[3] Fabian et al., "Adapt and diffuse: Sample-adaptive reconstruction via latent diffusion models." Proceedings of machine learning research 235 (2024): 12723.

---

> ### Author Rebuttal · Authors · 2025-07-29
>
> Thank you for your comprehensive review and constructive feedback. We address each concern below:
>
> * **Resolution** We appreciate this important concern about scalability. Kindly note that Appendix C2 reports experiments on 256x256 images. We will highlight these experiments in the main text.
>
> * **NFE Comparison** We have included NFE counts for all compared methods in Tables 1 and 2. Our method achieves superior performance while requiring significantly fewer NFEs than most competing approaches—a key practical advantage for real-world deployment. We will enhance the main text to better emphasize this computational efficiency.
>
> * **Additional Methods** We highlight that the cited works [1-3] train separate models for individual degradation types, whereas our approach trains a single unified model capable of handling multiple degradations. This represents a fundamentally different problem setting with distinct practical advantages. That said, we recognize the value of these comparisons, and we have trained and evaluated [2] on the multi-degradation task demonstrated in Table 1 (results below). Additionally, some approaches like [2] could potentially be combined with our architecture, which we see drastically improves the adaptation to the multi-degradation setting.
>
> | Method | Deblur | | | In-Painting | | | Matrix | | | NFEs |
> |-|-|-|-|-|-|-|-|-|-|-|
> |                | PSNR    | FID    | CFID    | PSNR    | FID    | CFID    | PSNR    | FID    | CFID    | |
> |InDI            | 21.4    | 21.6    | 22.8    | 17.7    | 22.6    | 25.0    | 13.7    | 35.5    | 58.7    | 50   |
> |InDI+InvFussion    | 22.9    | 6.4    | 6.4    | 18.5    | 8.4    | 8.5    | 16.7    | 14.9    | 15.2    | 50 |
> |InvFussion        | 22.7  | 4.2   | 4.2   | 17.1  | 5.0   | 5.0   | 15.1  | 7.1   | 7.1   | 63     |
>
> * **Degradation Aware** We appreciate this conceptual point and the reviewer's distinction. Any method attempting to restore from a zero-filled degradation $\boldsymbol{H}^\dagger\mathbf{y}$ is essentially operating in a blind setting, as no additional information is provided beyond the degraded sample (corresponding to Eq. (2)). In contrast, we use "degradation-aware" to denote methods that incorporate some—even partial—information from the known degradation, enabling a non-blind solution.\
> We agree with the reviewer that conditioning on the operator's action is not equivalent to conditioning on the operator itself. Nevertheless, our architecture gets to "probe" $\boldsymbol{H}$ several times along the network, each time applying it to different feature vectors. Therefore, theoretically, the network can deduce a lot about $\boldsymbol{H}$. Both our results and those of zero-shot methods (which also implicitly condition on the action of the operator) demonstrate that incorporating this partial information about the operator's action substantially improves results, as expected from the theory in Eq. (2).\
> While we maintain that our approach provides meaningful degradation conditioning beyond standard pseudo-inverse reconstruction, we will refine our positioning in the final version to more precisely describe our contribution within the existing framework of degradation-conditioned methods.
>
> * **LPIPS** We will include LPIPS scores for all experiments as requested. This will provide additional perspective on perceptual quality alongside our existing metrics. Here are the results of LPIPS on the experiments corresponding to Tables 1 and 2:
>
> |                | Deblur        || In-Painting        || Matrix            ||
> |-|-|-|-|-|-|-|
> |Dataset            | FFHQ     | ImageNet    | FFHQ    | ImageNet    | FFHQ | ImageNet    |
> |DDRM            | 0.095    | 0.246        | 0.268    | 0.458        | 0.517    | 0.277        |
> |DDNM               | 0.105    | 0.373        | 0.171    | 0.406        | 0.672    | 0.300        |
> |DPS             | 0.136    | 0.233        | 0.276    | 0.462        | 0.480    | 0.281        |
> |PiGDM             | 0.064    | 0.156        | 0.158    | 0.312        | 0.470    | 0.287        |
> |DAPS              | 0.683    | 0.736        | 0.148    | 0.327        | 0.535    | 0.263        |
> |Palette          | 0.068    | 0.147        | 0.103    | 0.215        | 0.409    | 0.236        |
> |InvFussion         | **0.053**    | **0.122**        | **0.101**    | **0.213**        | **0.340**    | **0.182**        |
> |MSE Methods   ||||||
> |Palette           | 0.126    | 0.327        | 0.144    | 0.318        | 0.606    | 0.314        |
> |InvFussion         | 0.106    | 0.278        | 0.143    | 0.315        | 0.469    | 0.237        |
>
> * **Cross-Attention** Thank you for this clarification. The reviewer is correct that "cross-attention" is the more standard terminology. Our "joint-attention" (also referred to as "extended-attention" in some literature) processes queries against both self-branch and cross-branch keys/values simultaneously, differing from the sequential "self-then-cross attention" approach. We will clarify this point in the final version. This architectural choice is based on our early experiments; we expect a “self-then-cross attention” architecture to perform nearly as well in all experiments.
>
> * **Typing Errors** We will address all noted typographical errors and consider updating the method name.
>
> * **Comparison to End-to-End Non-Diffusion Methods** We anticipate that diffusion-based approaches will demonstrate superior image quality compared to methods like Restormer, which employs transformer architecture (similar to ours) but is trained to minimize L1 loss. For scenarios requiring low distortion with reduced computational cost, our single NFE MMSE estimation method provides a viable alternative.
>
> * **Scaling With Respect to NFEs** We are not sure we understand the reviewer’s question. As a diffusion model, we expect image quality to worsen the fewer NFEs are used for sampling. Given the computational speedup compared to zero-shot methods, we have kept the NFE count as recommended by [28].

---

> > ### Comment · Reviewer_8Xy4 · 2025-08-05
> > **Response to authors**
> >
> > I thank the authors for their response and for pointing out that results on higher resolution images are already included in the appendix. I would recommend moving these to the main paper as I believe they are more meaningful. The authors also added results on InDI, which provides a relevant point for comparison. I encourage the authors to evaluate InDI on the other 2 datasets as well. Overall, I increase my score to reflect these improvements.

---

> > > ### Author Response · Authors · 2025-08-06
> > >
> > > We thank the reviewer for considering our rebuttal and for increasing their score. We will address the reviewer's suggestions in the final version.

---

### Official Review · Reviewer_Fzwc · 2025-07-08

**Clarity:** 3
**Significance:** 3
**Originality:** 3
**Rating:** 4
**Confidence:** 5

**Summary:**

The authors of this work propose a diffusion model based inverse problem solver that integrates measurement operators inside the underlying diffusion network and train it in that manner. This is done by applying the degradation operator, a linear transformation, and pseudoinverse of the degradation operator to rearranged (in shape of stack of images) deep features. Results are shown for some image restoration tasks on standard datasets.

**Questions:**

1.	What are the PSNRs of the “not seen in training” results in Figure 3? How do they compare to the PSNRs in Table 3?

2.	In Section 4.1, the authors can explain unconditional generation better.  The MMSE predictor model could be explained better.

3.	The authors state “Projection operators are not well-defined for noisy degradations” – which is not clear. Projection could also be done onto the set ||y-Hx|| < epsilon with epsilon set based on the noise. Have the authors compared to other recent diffusion models that incorporate measurement noise information?

**Ethical Concerns:**

["NO or VERY MINOR ethics concerns only"]

**Final Justification:**

The authors responded to my comments and promised to do some more updates in final version. Based on the replies, I increased my score to reflect my opinion of the work.

**Limitations:**

The authors oversell their approach at numerous places.

**Quality:**

3

**Strengths And Weaknesses:**

Strengths

1.	The approach offers a novel way to integrate measurement operators into the diffusion model without running into the computational cost of explicitly passing them as network inputs.

2.	Results are shown for a few tasks comparing to training-based and zero-short (i.e., measurement operator is integrated at inference time) diffusion methods.

Weaknesses

1.	The approach is novel but ad hoc/ill-motivated without the solid basis in physical consistency that many other image reconstruction models like unrolled networks deploy.

2.	The approach requires the deep features to match the image dimension (or be modified for it) – to apply the degradation operator – something not motivated well.

3.	In several imaging systems, H and its pseudo-inverse can be expensive to apply rendering the training and application of the diffusion model expensive.

4.	The authors mention unrolling methods but the results section lacks comparison to state of the art physics-based supervised unrolled networks as well as recent zero-shot deep image prior (DIP), etc., or combined diffusion-DIP methods.

5.	The PSNRs in Table 3 are quite poor. Perhaps, other kinds of out of training degradation testing could be shown for more comprehensive study.

6.	InvFussion (without MMSE predictor) offers worse performance in several situations than other zero-shot methods compared with. This lacks more detailed analysis.

7.	The comparisons to diffusion models is quite limited as there are more recent methods with much better performance than DAPS, etc.

8.	There are several typos/English grammar/spelling errors in the manuscript. Please proofread. “preform”, “Bayse”, etc.

---

> ### Author Rebuttal · Authors · 2025-07-29
>
> Thank you for your comprehensive review and constructive feedback. We address each concern below:
>
> * **Motivation** Rather than enforcing physical consistency through explicit constraints like unrolling methods, our approach is motivated by the principle of learning-based adaptation to measurement operators. Even though consistency with the measurements is not enforced, we find that our architecture manages to learn to satisfy the consistency constraint quite well.
>
> * **Deep Feature Dimensions** This architectural constraint is readily accommodated in modern diffusion architectures, particularly DiT variants, through straightforward hyperparameter adjustments that do not compromise performance. The implementation involves rounding the hidden dimension in specific layers to ensure the total number of hidden feature elements is divisible by the total image dimensions. Importantly, this constraint need not apply to all layers, enabling architectural flexibility. We will clarify the scope and implementation details of this limitation in the final version.
>
> * **Computation of $\boldsymbol{H}$** While we acknowledge this computational consideration, it is important to emphasize that this limitation is inherent to many inverse problem solving methods, including established zero-shot approaches. Moreover, the practical applications we target—and many others in the field—benefit from efficient implementations of both $\boldsymbol{H}$ and $\boldsymbol{H}^\dagger$. We will explicitly discuss this computational consideration in the final version.
>
> * **Comparison to Unrolling** While we agree that comparisons to supervised unrolled networks would provide valuable context, direct comparison presents significant practical challenges for our evaluation framework. Methods like Deep Image Prior operate at the single-image level, making a single 10K FID evaluation computationally prohibitive. Additionally, these methods typically optimize for different objectives—such as MAP estimation rather than posterior sampling quality—and employ fundamentally different architectures, limiting the conclusiveness of direct comparisons.
> To provide meaningful context nonetheless, we evaluated PSNR and LPIPS metrics for approximately 250 images restored using ASeq-DIP [4], a recent DIP method. \
> Despite ASeq-DIP's significantly higher per-sample computational cost, InvFussion achieves superior performance across all metrics. We will include this analysis in the final manuscript.
>
> |Method        | Deblur    || In-Painting    || Matrix         ||
> |-----------|-------|-------|-------|-------|-------|-------|
> |            | PSNR    | LPIPS    | PSNR    | LPIPS    | PSNR    | LPIPS    |
> |ASeq-DIP [4]    | 22.6    | 0.423    | 12.9    | 0.660    | 12.6    | 0.924    |
> |InvFussion    | 23.4    | 0.278    | 18.4    | 0.315    | 17.3    | 0.469    |
>
> * **Table 3** We acknowledge the reviewer's concern regarding the PSNR values. To address this, we repeat the experiment with a larger constant out-painting mask, yielding the following results below.\
> These results maintain the same performance trends observed in Table 3. We emphasize that our primary objective is robust performance on degradations *within the training distribution*, which encompasses a wide and diverse range of conditions.
>
> |Method	| PSNR	| FID		| CFID		|
> |-|-|-|-|
> |DDRM		| 14.9		| 9.5		| 9.5		|
> |DDNM          	| 17.0		| 21.6		| 21.6		|
> |DPS               	| 11.3		| 10.7		| 15.8		|
> |PiGDM           	| 15.6		| 3.6		| 3.6		|
> |DAPS             	| 16.6		| 12.8		| 10.0		|
> |Palette		| 16.0		| 7.6		| 5.7		|
> |InvFussion      | 16.7		| 3.0		| 3.0		|
> |MSE Methods |||
> |Palette		| 18.2		| -		| -		|
> |InvFussion      | 19.5		| -		| -		|
>
> * **PSNR Comparison** The PSNRs reflect the inherent Perception Distortion Tradeoff [1] in restoration methods. Our approach prioritizes generating perceptually realistic samples, as evidenced by consistently superior FID scores, while maintaining competitive PSNR values among methods with comparable perceptual quality. The MMSE variant demonstrates our model's flexibility to achieve higher PSNR when distortion minimization is the primary objective
> * **Other Methods** In our work, we have included comparisons with a representative selection of well-established zero-shot methods that remain dominant in the field. Following suggestions from reviewers 8Xy4 and XFfT, we will incorporate additional comparisons with InDI [2] (training-based) and MGPS [3] (zero-shot), both representing more recent developments. These comparisons demonstrate competitive performance for our proposed method.
>
> |Method   | Deblur   ||| In-Painting      ||| Matrix ||| NFEs |
> |-|-|-|-|-|-|-|-|-|-|-|
> |        | PSNR    | FID    | CFID    | PSNR    | FID    | CFID    | PSNR    | FID    | CFID    |
> |MGPS [3]   | 20.5    | 7.7    | 9.7    | 13.9    | 10.5    | 24.3    | 11.7    | 8.2    | 29.0    | 1764 |
> |InDI [2]           | 21.4    | 21.6    | 22.8    | 17.7    | 22.6    | 25.0    | 13.7    | 35.5    | 58.7    | 50   |
> |InvFussion | 22.7    | 4.2    | 4.2    | 17.1    | 5.0    | 5.0    | 15.1    | 7.1    | 7.1    | 63	 |
>
> * **Typos** We will conduct thorough proofreading to address all typographical and grammatical errors in the final manuscript.
>
> * **PSNR of examples** For the facial image example shown in Figure 3: DDRM: 8.98 dB, DDNM: 11.06 dB, DPS: 13.14 dB, PGDM: 14.85 dB, DAPS: 9.44 dB, Palette: 13.65 dB, InvFussion: 16.06 dB. These values align with the trends observed in Table 3, demonstrating consistent performance across different test scenarios.
>
> * **Unconditional and MMSE** We will improve our explanations of these concepts in the final version. For unconditional generation, we simply omit conditioning on any measurement, allowing the model to generate samples from the prior distribution rather than the posterior distribution given a degraded image. For MMSE estimation, we demonstrate how training-based inverse problem solvers can be adapted for MMSE estimation by leveraging the near-equivalence between the high-noise regime in diffusion training and linear regression problems.
>
> * **Projection for Noisy Degradations** The $\epsilon$-ball projection suggested by the reviewer ($||\mathbf{y}-\boldsymbol{H}\mathbf{x}|| < \epsilon$) can indeed be used as a projection. The CFID values with this projection, corresponding to Table 8, are as follows:
>
> |                | Deblur        | In-Painting        | Matrix            |
> |-|-|-|-|
> |DDRM            | 96.9   | 124.7        | 85.4    |
> |DDNM               | 59.5    | 75.2       | 137.9   |
> |DPS             | 12.3    | 20.2       | 25.5    |
> |PiGDM             | 8.7    | 23.8        | 19.2   |
> |DAPS              | 78.1    | 61.9        | 75.7    |
> |Palette          | 4.7    | 5.3       | 6.2    |
> |InvFussion         | **4.3**    | **5.0**        | **6.2**    |
>
> [1] Blau, Yochai, and Tomer Michaeli. "The perception-distortion tradeoff." Proceedings of the IEEE conference on computer vision and pattern recognition (2018).
>
> [2] Delbracio et al., "Inversion by direct iteration: An alternative to denoising diffusion for image restoration." arXiv preprint arXiv:2303.11435 (2023).
>
> [3] Moufad, Badr, et al. "Variational Diffusion Posterior Sampling with Midpoint Guidance." The Thirteenth International Conference on Learning Representations (2025).
>
> [4] Alkhouri, Ismail, et al. "Image reconstruction via autoencoding sequential deep image prior." Advances in Neural Information Processing Systems 37 (2024).

---

> > ### Comment · Reviewer_Fzwc · 2025-08-06
> >
> > Thanks to the authors for their added responses to comments. I have some follow-ups.
> >
> > Regarding the ad hoc nature of the approach they say “Even though consistency with the measurements is not enforced, we find that our architecture manages to learn to satisfy the consistency constraint quite well.” – this needs to be unpacked better in the reply. Perhaps, the authors can do more ablation studies to explore this question better although a theoretical understanding would be ideal. The match between FID and CFID scores for the proposed method can also be expanded on.
> >
> > I didn’t quite get the response on comparing to physics-based or unrolling methods “While we agree that comparisons to supervised unrolled networks would provide valuable context, direct comparison presents significant practical challenges for our evaluation framework.” I still think this is an important comparison as the goal is to solve inverse problems here not just improve one method type. The comparison to a DIP method is appreciated. However, it is missing details of how the hyperparameters or settings of the compared method were set or optimized to ensure fair comparisons against a fully data-free method. The other experiments added in response to comments are good. I am willing to reconsider my score with  additional input on above points.

---

> ### Author Response · Authors · 2025-08-08
>
> We thank the reviewer for considering our rebuttal and for their continued engagement.
>
> **Consistency** Zero-shot methods — and indeed data-free DIP methods — rely on explicit closed-form mappings between measurements $\mathbf{y}$ and the operator $\boldsymbol{H}$. In contrast, InvFussion employs supervised training to learn this mapping implicitly through the training loss, rather than enforcing consistency explicitly at test time.
>
> From a theoretical perspective, this implicit consistency can be understood through our network's optimal denoising solution, $\mathbb{E}[\mathbf{x}_0 | \mathbf{x}_t, t, \mathbf{y}, \boldsymbol{H}]$. This expectation, which is used in the generation process, is inherently consistent with the measurements because it conditions on both $\mathbf{y}$ and $\boldsymbol{H}$. Crucially, without the explicit conditioning on $\boldsymbol{H}$ enabled by InvFussion, the standard optimal solution $\mathbb{E}[\mathbf{x}_0 | \mathbf{x}_t, t, \mathbf{y}]$ lacks this consistency guarantee. This is because without accounting for the specific degradation process, the expectation also averages all possible degradations, similar to Eq. (2) in our paper.
>
> This theoretical consistency is validated by our results: InvFusion's CFID scores are nearly identical to the corresponding FID values, indicating that projecting our model's outputs onto the measurement space has minimal impact. This suggests that our network has indeed learned to produce outputs that are already highly consistent with the measurements. To provide more direct evidence of this consistency, we include quantitative measurements of MSE in the degradation's range-space below, corresponding to the experiment in Table 2. MSE is used to accommodate degradations of differing rank and range, and therefore has a different scale for each degradation class. The MSE values confirm that InvFussion achieves consistency comparable to methods that employ explicit projections, such as DDNM or DDRM. These results offer a complementary metric to the FID/CFID analysis.
>
> |Method             |Deblur        |In-Painting|Matrix        |
> |----------------|----------|-----------|-----------|
> |DDRM               |6.90        | 0.00        | 0.00        |
> |DDNM            |2.28        | 0.00        | 0.00        |
> |DPS                |789.63    | 10.27        | 0.00        |
> |PiGDM              |3.22        | 0.00        | 0.00        |
> |DAPS               |720765.61    | 2.89        | 5.68        |
> |MGPS               |2087.28    | 23.55        | 9.30        |
> |Palette            |115.67    | 0.23        | 11.57        |
> |InvFussion         |5.94        | 0.01        | 0.00        |
>
> Regarding ablation studies, we propose to include an ablation study examining the effect of varying the number of layers that utilize the InvFussion block. This will help quantify how repeated exposure to the degradation operator influences the final output consistency and provide deeper insights into our architectural choices.
>
> **Comparison to Unrolling and DIP** The challenge raised by supervised unrolling methods in our setting is the adaptation of the training scheme of such approaches to a multi-degradation framework. While we cannot complete this within the scope of the current rebuttal time frame, we commit to including results on supervised unrolling methods in the final version of our manuscript.
>
> Regarding hyperparameter settings for ASeq-DIP, we employed the default parameters from the official GitHub implementation; UNet architecture with two levels, Adam optimizer with a learning rate of 1e-4, 5000 outer training iterations and 10 inner ones. We have not been able to explore hyperparameter settings within the rebuttal time frame, because a separate network must be optimized for each sample, leading to a high computational cost and an extremely slow runtime. We will include a more thorough comparison in the final revision where we will test different hyperparameters for ASeq-DIP. At the same time, to ensure a fair comparison we must take into account the tradeoff between costly hyperparameters and total algorithm runtime, as our method specifically aims to alleviate the longer run-times of zero-shot methods.
>
> We are happy to respond to additional questions the reviewer may have.

---

> > ### Comment · Reviewer_Fzwc · 2025-08-08
> >
> > Thanks for the responses. The MSE study and comment on doing more thorough comparisons is reasonable. So I will increase my score accordingly given this context for revisions.

---

### Comment · Area_Chair_J8no · 2025-08-03

Dear authors and reviewers,

First of all, thank you all for your efforts so far. The author-reviewer discussion period will end on August 6.

@Authors: If not done already, please answer all questions raised by the reviewers. Remain factual, short and concise in your responses, and make sure to address all points raised.

@Reviewers: Read the authors' responses and further discuss the paper with the authors if necessary. In particular, if the concerns you raised have been addressed, take the opportunity to update your review and score accordingly. If some concerns remain, or if you share concerns raised by other reviewers, please make sure to clearly state them in your review. In this case, consider updating your review accordingly (positively or negatively). You can also maintain your review as is, if you feel that the authors' responses did not address your concerns.

I will reach out to you again during the reviewer-AC discussion period (August 7 to August 13) to finalize the reviews and scores.

The AC

---

> ### Comment · Area_Chair_J8no · 2025-08-08
>
> Dear reviewers,
>
> The reviewers-authors discussion phase will end in less than 24 hours.
>
> If not done already, make sure to submit the "Mandatory Acknowledgement" that confirms that you have read the reviews, participated in the discussion, and provided final feedback in the "Final justification" text box.
>
> Be mindful of the time and efforts the authors have invested in answering your questions and at least acknowledge their responses. Make sure to provide a fair and scientifically grounded review and score. If you have changed your mind about the paper, please update your review and score accordingly. If you have not changed your mind, please provide a clear and sound justification for your final review and score.
>
> Best regards,
> The AC

---

### Note · Authors · 2025-08-11

We thank the reviewers and the Area Chair for their attention and engagement.

For the convenience of the reviewers and AC, we summarize the key points raised and our rebuttal.

**Experimental Comparisons and Evaluation:** Multiple reviewers (Fzwc, 8Xy4, XFfT) requested additional comparisons to recent methods. We provided evaluations against InDI, MGPS, and ASeq-DIP, demonstrating superior performance while requiring similar or fewer NFEs. Reviewers expressed satisfaction with these additions, requesting we complete these assessments for other experiments in our final version. We also added LPIPS scores as requested by Reviewer 8Xy4, where our method leads. Reviewer XFfT specifically noted the quality of our empirical results.

**Theoretical Foundation and Consistency:** Reviewer Fzwc inquired about the method's theoretical basis and underlying motivation. We provided a theoretical analysis showing how our network learns implicit measurement consistency through supervised training, supported by MSE consistency metrics demonstrating performance comparable to methods with explicit projections. The reviewer indicated satisfaction with our response.

**Resolution and Computational Efficiency:** Reviewer 8Xy4 raised concerns about low-resolution experiments and computational costs. We highlighted existing 256×256 results in our appendix as well as InvFussion's significant NFE advantages over competing methods. The reviewer acknowledged these points and recommended moving higher-resolution results to the main text.

**Ablation Studies:** We responded to concerns about ablation studies with results for experiments suggested by Reviewer 21uD and will conduct additional ablations which would address consistency concerns raised by Reviewer Fzwc.

**Originality:** All reviewers have indicated that our method is original, with Reviewers XFfT and 21uD specifically stating this as a strength of the approach.

**Motivation and Generalization:** Reviewer 21uD maintained concerns about limited generalization compared to zero-shot methods. While acknowledging this limitation, we demonstrated strong within-family generalization (e.g., random matrix experiments) and highlighted practical advantages including computational efficiency and enabling downstream applications like uncertainty quantification.

We hope the reviewers are in agreement with this summary and will positively consider our work. Our final version will incorporate rebuttal experiments and discussions.

---

### Decision · Program_Chairs · 2025-09-17

**Decision:**

Accept (poster)

**Comment:**

The average rating is 4, with reviewers mostly recommending acceptance (4, 3, 4, 5). The discussion has been constructive and improvements have been made to the paper, to the satisfaction of most reviewers. Concerns remain about the significance of the contribution. Given the otherwise positive reviews and the constructive discussion, I tend to recommend acceptance, as significance is often difficult to assess a priori. I encourage the authors to address the reviewers' comments and to implement the modifications discussed with the reviewers in the final version of the paper.